



# Tomographic retrieval algorithm of OH concentration profiles using Double Spatial Heterodyne Spectrometers

Yuan An[1,2,3], Jinji Ma[1,2,3], Yibo Gao[1,2,3], Wei Xiong[4,5], Xianhua Wang[4,5]

[1] School of Geography and Tourism, Anhui Normal University, Wuhu, 241003, China

[2] Engineering Technology Research Center of Resource Environment and GIS, Wuhu, 241003, China

[3] Key Laboratory of the Research of Natural Disaster Process and Prevention, Wuhu, 241003, China

[4] Anhui Institute of Optics and Fine Mechanics, Hefei Institutes of Physical Science, Chinese Academy of Sciences, Hefei, 230031, China

[5] Key Laboratory of Optical Calibration and Characterization of Chinese Academy of Sciences, Hefei, 230031, China

**Correspondence:** Jinji Ma (jinjima@ahnu.edu.cn)

**Abstract.** The hydroxyl radical (OH) determines the atmospheric self-cleaning capability and is one of the significant oxidants in atmospheric photochemistry reactions. The global OH has been monitored by satellites with the traditional limb mode in the past decades. This observed mode can achieve high-resolution vertical OH data, but cannot obtain the enough horizontal OH data for inverting high-precision OH concentrations because OH has the high reactivity that makes its concentrations extremely low and distributions complicated. The Double Spatial Heterodyne Spectrometers (DSHS) is designed in order to achieve more high-resolution and detailed OH data. This sensor can measure the OH by the three-dimensional limb mode to obtain the comprehensive OH data in the atmosphere. We propose a new tomographic retrieval algorithm here based on the simulated observation data due to the DSHS will work officially on the orbit in the future. We build up an accurate forward model that the main part of it is the SCIATRAN radiative transfer model which is modified according to the radiation transmission theory. We also construct the tomographic retrieval algorithm that the core is a look up table method. A tomographic observed database is built up through the atmospheric model, the spatial information (position of the target area and satellite position), the date parameters, the observation geometries, OH concentrations and simulated observation data. The OH concentrations can be found directly from it. If there are no corresponding query conditions in the tomographic observed database, the cubic spline interpolation is used to obtain the OH concentrations. The tomographic retrieval algorithm can obtain the more accurate OH concentrations even in the lower atmosphere where the OH data is not well and avoids the initial guess values for solving the iteration problems. Our research not only provides a scientific theory support for the construction of DSHS, but also gives a new retrieval algorithm idea for other radicals.

# 1 Introduction

The OH plays an important and initial role in atmospheric photochemistry reactions because of its strong oxidation It



can remove many natural and anthropogenic compositions which are significant for the air quality, the ozone distributions even the climate change from the atmosphere (Stevens et al., 1994). OH mainly derives from the reactions of O($^1$D), a

photolytic product of ozone in the ultraviolet band, with water vapor in the middle and upper atmosphere. The removal of OH is affected by various compounds like nitrogen oxides, sulfur dioxide, carbon monoxide, methane and other volatile organic compounds (Lelieveld et al., 2016;Wolfe et al., 2019;Zhang et al., 2018). Therefore, on the one hand, the OH affects the photochemical and kinetic processes in the atmosphere and reflects the short-term and long-term climatic evolution processes in some respects. On the other hand, it is of great significance to enhance the understanding

of the atmospheric physical and chemical processes.

It is a great challenge to monitor the OH in the atmosphere because of its low concentrations and strong activity. The Fluorescence Assay by Gas Expansion (FAGE), the Differential Optical Absorption Spectroscopy (DOAS) and the Chemical Ionization Mass Spectrometry (CIMS) are commonly used to measure the OH concentrations in the actual limited environments (Hard et al., 1984;Mauldin et al., 1998;Perner et al., 1976). In addition, the $^{14}$CO oxidation method,

the Scrubbing using the salicylic acid Technique and the Spin Trapping method are used to obtain the OH concentrations in the laboratory for some theoretical researches (Felton et al., 1990;Salmon et al., 2004;Watanabe et al., 1982). Apart from the six physical and chemical methods mentioned above, many researchers used the high-precision spectrum data from the ground-based instruments especially the Fourier Transform Ultraviolet Spectrometer (FTUVS) in the Table Mountain (Cageao et al., 2001;Cheung et al., 2008;Mills et al., 2002). These methods are restricted by the features of

the instruments which can provide accurate OH data in the limited lower atmosphere but are difficult to provide enough data for inverting OH concentrations from mesopause to tropopause. The development of satellite technologies carries out the global wide-scale detection of OH. Conway retrieved OH concentrations by the least squares fitting method until the success of the Middle Atmosphere High-Resolution Spectrometer Investigation (MAHRSI) which provided the first observed data of OH in the mesosphere with diffractions grating technology from space (Conway et al., 1999). A new

interferometric technique called spatial heterodyne spectroscopy was applied to the Spatial Heterodyne Imager for Mesospheric Radicals (SHIMMER) for reaching the higher spectral resolution on the space shuttle middeck with small size and no mobile optical components. This satellite sensor measured the OH solar resonance fluorescence by the limb mode from the low Earth orbit to obtain more accurate OH data. Englert used the same method as the MAHRSI to invert OH concentrations (Englert et al., 2008). Besides these sensors mentioned above, the 2.5-THz radiometer on the

Microwave Limb Sounder (MLS) was designed to measure the thermal emission signal of OH from stratosphere to mesosphere because the spectral region around the pair of strong OH lines at 2.51 and 2.514 THz is clean relatively. The sensitive thermal emission data is used to invert the volume mixing ratio of OH concentrations, however the results have a poor signal to noise ratio in some atmospheric regions. Livesey used the standard optimal estimation method to obtain the OH concentrations from the calibrated MLS observed data (Level 1B) with the radiative transfer equation. He

also corrected the noisy products by a separate task using the full radiance dataset as the relevant band after the main process (Livesey et al., 2006). Wang used the average orthogonal fitting method to handle with the MLS data for obtaining the OH concentrations, and compared the results with the ground-based FTUVS results in some areas. They matched very well (Wang et al., 2008). Damiani handled the MLS data of nighttime winter in the northern hemisphere from the latitude of 75° to 82° by regional mean method to research the change of OH concentrations in short-term (day)



and long-term (weeks) dimension (Damiani et al., 2010).

The MAHRSI and SHIMMER measured the OH solar resonance fluorescence in the $A^2\Sigma^+ - X^2\Pi(0,0)$ band at the wavelength around 309 nm and proved that is the more possible and effective way to monitor atmospheric OH from space load at present. The traditional limb mode can obtain high-resolution vertical OH radiance profile which contains the information of OH concentrations at different tangent plane because of the benefit of spectral technology. However,

the OH radiance which is measured at one field of view is the sum of the overall contributions in the corresponding fields of view, but the OH is inhomogeneous in fact. These will lead to some precision errors (Englert et al., 2010). Anhui Institute of Optics and Fine Mechanics, Chinese Academy of Sciences, has done some researches on the DSHS in advance to improve the horizontal resolution of OH data and achieve the better structure of OH profiles. The DSHS, will work on the orbit at 500 km, is designed to measure the OH solar resonance fluorescence from 15 to 85 km by using the

double spatial heterodyne spectrometers in orthogonal layout. The field of view angle is designed as 2 degree and the spectral resolution will reach 0.02 nm at least. This paper gives the OH interferogram which is emulated by a forward model and describes a new method called the tomographic retrieval algorithm which uses simulated DSHS data and theory of the look up table method to invert OH concentrations. The OH concentrations in the target area will be found out in the tomographic observed database which constructs by the atmospheric model, the spatial information, the date

parameters, the observation geometries, OH concentrations productions and the simulated observation data. The retrieval algorithm can obtain more accurate OH concentrations in the middle and upper atmosphere especially in the lower atmosphere than the results which are obtained by traditional limb retrieval algorithms and can raise the inversion speed compared with the traditional limb retrieval algorithms.

# 2 Measurement strategy

## 2.1 The principle of spatial heterodyne spectroscopy


The DSHS is designed based on the principle of the spatial heterodyne spectrometer which the optical structure is shown in Fig. 1. The two planar mirrors in the traditional Michelson interferometer are replaced by the two diffraction gratings (grating 1, grating 2). The light, enters through aperture A, is collimated by the lens L1 and is split into two beams of coherent light of equal intensity by the beam splitter. One of the them is reflected by the beam splitter and incidents on

the grating 1. The light is diffracted by the grating 1 and returns to the beam splitter. The other light incidents on the grating 2 through the beam splitter as well. It is diffracted by the grating 2 and reflects back to the beam splitter. The gratings are fixed which are placed at a Littrow angle $\theta$ with the orthogonal plane of the optical axis in the spatial heterodyne spectrometer system. The light incidents on the gratings by $\theta$ angle and diffracts back with $\theta$ angle at some wavenumbers. It is called the light of Littrow wavenumber $\sigma_0$. The two exiting wavefronts of light of $\sigma_0$ are diffracted

by the gratings and perpendicular to the optical axis. The phase difference is zero and the interference fringe spatial frequency is zero that cannot form the interferogram. However, the light of non-Littrow wavenumber has an $\pm\gamma$ angle





with the optical axis. The light forms interferogram which will image on the imaging detector by the optical imaging systems L2 and L3. So, the overlapping two wave surfaces have an angle of $2\gamma$ which are calculated by the Eq. (1) (Dohi and Suzuki, 1971):


$$\sigma[\sin\theta + \sin(\theta - \gamma)] = \frac{m}{d} \tag{1}$$

where the $\sigma$ is the wavenumber of light, $m$ is the order of diffraction, $1/d$ is the grating groove density. The angle between the light of arbitrary wavenumber and the exiting light of Littrow wavenumber is $\gamma$. The spatial frequency of

two $\sigma$ light is given by the Eq. (2):

$$f_x = 2\sigma \sin\gamma \approx 4(\sigma - \sigma_0)\tan\theta \tag{2}$$

For the input spectrum $B(\sigma)$, the intensity on the imaging detector is given by Eq. (3):


$$I(x) = \int_0^\infty B(\sigma)\big(1 + \cos\big(2\pi(4(\sigma - \sigma_0)x\tan\theta)\big)\big)d\sigma \tag{3}$$

The spectral curve $B(\sigma)$ can recover from interferogram $I(x)$ by Fourier transform algorithm.

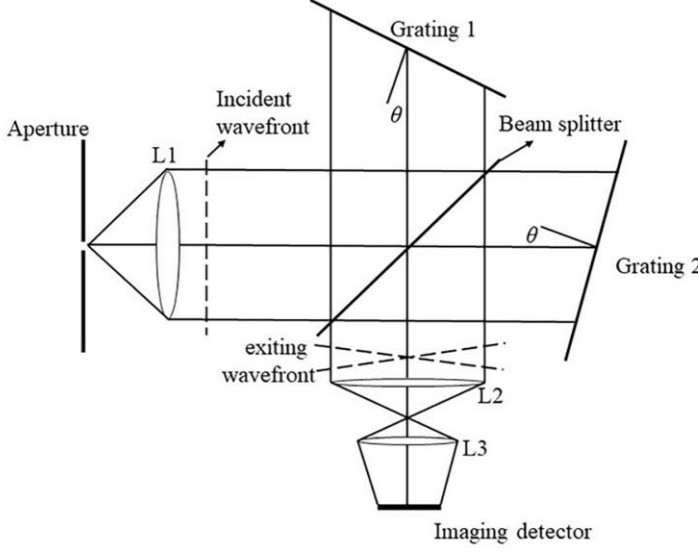


**Figure 1.** Schematic diagram of spatial heterodyne spectrometer.



Furthermore, the spatial heterodyne spectroscopy can acquire spatial distribution of one-dimensional spectral information through the two-dimensional detection technology. The scene in the field of view is divided into multiple field of view slices which are less or equal to the number of rows in the focal plane array by adding a cylindrical mirror to the front or rear optical system. The interferograms of each field of view slices are imaged on the corresponding detector rows respectively. The several rows on the detector correspond to a layered spectral information in a field of view. The corresponding spatial resolution unit on the detector can indicate the spatial information when the target signal appears in a certain range of line. These mean that the spatial heterodyne spectroscopy can obtain the radiance of atmosphere information at different altitudes simultaneously without scanning the atmosphere from a height to other height respectively. It is important for monitoring the OH to reduce the time of data acquisition because the OH is variable fast in the spatial and temporal dimension.

## 2.2 Instrument innovation

The spatial heterodyne spectroscopy technique, an ultra-high-resolution spectroscopy technique used in the DSHS, has the characteristics of high flux, no mobile part and small size. The DSHS which is shown in the Fig. 2 mainly consists of two spatial heterodyne spectrometers in the orthogonal layout to obtain the enough data for inverting the OH concentrations. It includes the telescope cylinder system, the collimating system, the information processing system, the imaging systems and some optical components else. One of the spatial heterodyne spectrometers which scans along the satellite working orbit is defined as SHS1, the other which scans across the satellite working orbit is defined as SHS2. The scanning direction of SHS2 is orthogonal to SHS1s' due to the special design of the orthogonal layout. The hierarchical detection of a series of observed radiance in the same target area comes true through the movement of satellite platform.

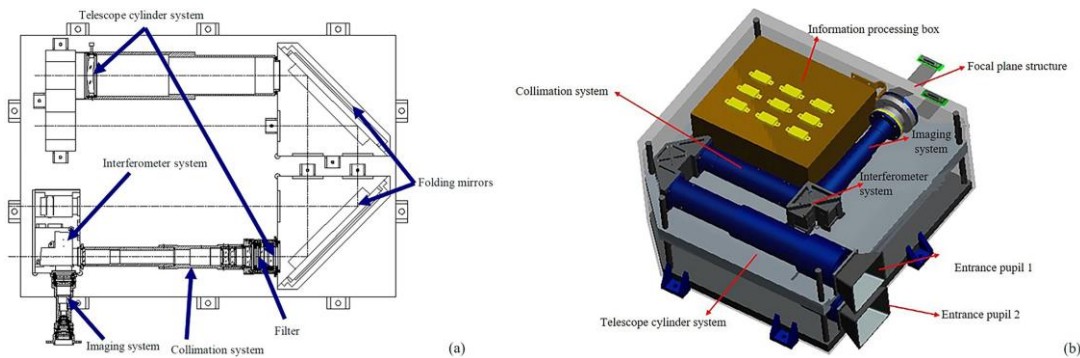

**Figure 2.** Illustration of the Double Spatial Heterodyne Spectrometers. the left part (a) shows the vertical view of the DSHS and the right part (b) shows the overall mold of the DSHS.



The DSHS will detect the altitudes from 15 to 85 km by the three-dimensional limb mode. It can obtain the observed data in the target area along the satellite working orbit and across the satellite working orbit respectively at the same time because the SHS1 and SHS2 are orthogonal to each other. The Figure 3 shows how the tomographic data is obtained in the process of satellite working.

As the Fig. 3 (a) shows that there are no intersection regions between the field of view slices of SHS1 and SHS2s' at the
time of T1. The tomographic data cannot be obtained in this situation. The three-dimensional structure of OH data is unable to be reconstructed. The data of SHS1 and SHS2 at different time will be used to get an intersection region for solving this issue. The different time which meet the requirements of three-dimensional limb mode of the DSHS (we define the first time as the time of T1 and the next time as the time of T2) is shown in the Fig. 3 (b). At the time of T1, the SHS1 and SHS2 finish a limb scanning in the vertical dimension (along the satellite working orbit) and the horizontal
dimension (across the satellite working orbit), then the satellite platform will work into the next position at the time of T2 and finishes the next observation project. The line of sight at the time of T1 will intersect with the line of sight at the time of T2. The intersection region (target area) will form. However, the composition of tomographic data does not need all data from the SHS1 and SHS2 at the time of T1 and T2. Some data should be omitted. As the Fig. 3 (c) shows that the field of view slices of T2-SHS2 (a horizontal dimension spectrometer at the time of T2) are orthogonal to the field
of view slices of T1-SHS1 (a vertical dimension spectrometer at the time of T1) in an intersection region. The data obtained by the SHS2 at the time of T1 and the SHS1 at the time of T2 will be omitted. Thus, the three-dimensional segmentation of the target area will complete. The three-dimensional division of the target atmosphere which consists of the intersection regions of SHS1 and SHS2 at different time is accomplished finally with the satellite platform moving forward continually. Different combinations of data for different intersection regions from two spatial heterodyne
spectrometers at different time are used to invert OH concentrations.





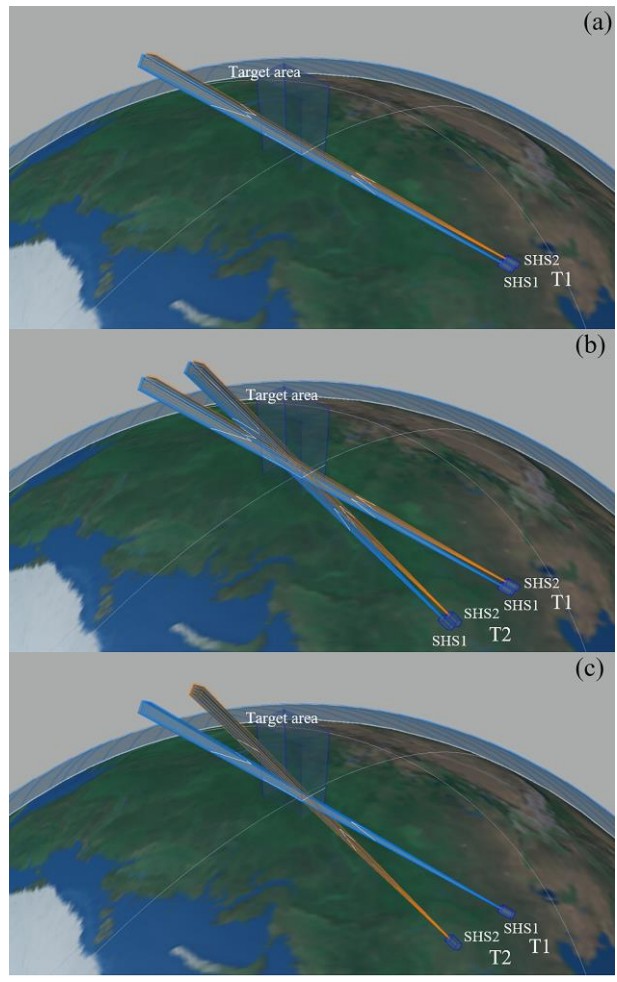

**Figure 3.** Three-dimensional limb mode of DSHS at the time of T1 and T2 (a). The DSHS consists of two spatial heterodyne spectrometers (SHS1 and SHS2). The data of target area cannot be obtained by DSHS at the time of T1 and T2 respectively (b). DSHS monitors the OH in the target area using the data of SHS1 at the time of T1 and SHS2 at the time of T2 together (c).

The scanning direction will not be completely vertical or parallel to the surface when the DSHS will be on the orbit, but will have an angle η. So, the satellite positions at the time of T2 where T2-SHS2 is orthogonal to T1-SHS1 in the intersection region are different with the η changing. In order to display and descript the satellite position of T2 conveniently, the η is set to 0° temporarily. However, a unified set of η is 30° in the relevant calculation during pre-research phase.

The satellite positions and observation geometries are given at the time of T1. The satellite positions at the time of T2 need to calculate for realizing the three-dimensional segmentation of the target atmosphere. It indicates that there are





two satellite positions where the scanning directions of T2-SHS2 and T1-SHS1 are orthogonal to each other in the intersection region at the time of T2 based on the theory mentioned above. Figure 4 shows that there is a tangent point which is defined as P1 when the DSHS detects the atmosphere by the three-dimensional limb mode. The intersection region along the lines of sight of SHS1 (the blue line in the Fig. 4) and SHS2 are the observed data at the time of T1.

There will be two lines of sight (the red lines in the Fig. 4) where the scanning direction of SHS2 is orthogonal to the SHS1s' under the premise that the working altitude of the satellite is constant. The two lines of sight at the time of T2 (T2-1 and T2-2) and a single line of sight at the time of T1 form two intersection regions R1 and R2 which are symmetrical about P1 along the line of sight at the time of T1. The distances between the intersection region and the tangent point determine the spatial resolution of the three-dimensional limb mode. That means, the smaller distance

between the P1 and R1 (or R2) is, the higher spatial resolution will be.

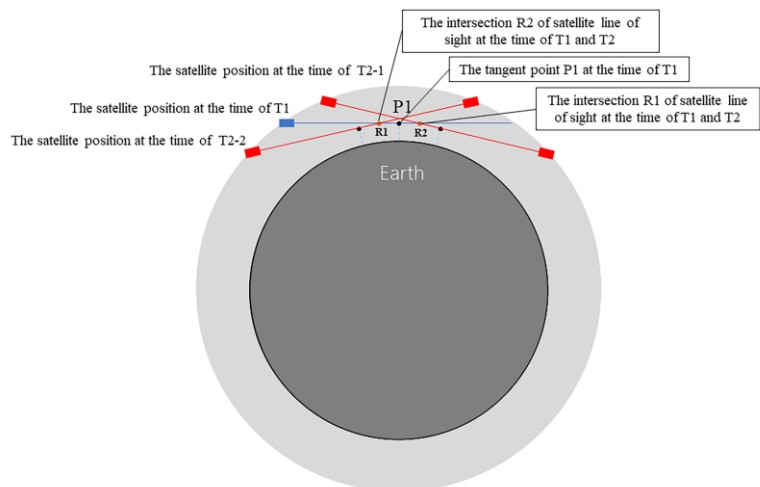

**Figure 4.** Limb geometry diagram for the three-dimensional limb mode at the time of T1 and T2. The blue line is the
line of sight of SHS1. The red lines are the lines of sight of SHS2.

Figure 5 shows relationship of the satellite positions at the time of T1 and T2 in the space. These two intersection regions are on the two sides of tangent point and are at the altitude of 500 km. The satellite position on the same side to the time of T1 is defined as the position at the time of T2-1, and on the opposite side to the time of T1 is defined as the position

at the time of T2-2. The distance of the position at the time of T2-2 from the time of T1 is farther than the position at the time of T2-1 that means satellite needs more time to work from the position of the time of T1 to the position at the time of T2-2. The observed data will change a lot in this process and cannot reflect the actual OH distributions in the target area at the time of T1. The satellite position at the time of T2-2 should be omitted. However, the satellite position at the time of T2-1 is near the position at the time of T1, the observed data at the position of the time of T2-1 can reflect the





OH distribution in the target area better because the OH information does not change too much in the short working time
       of satellite.

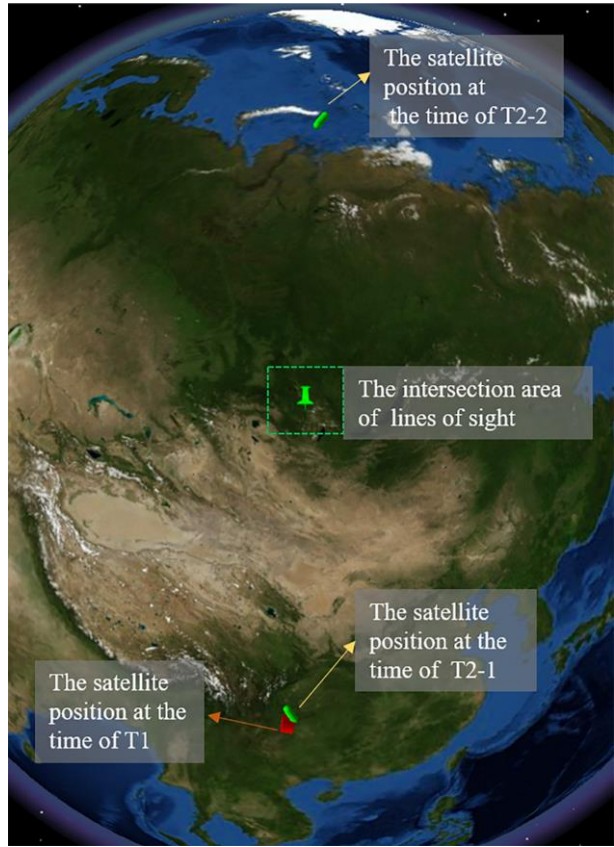

**Figure 5.** Relative location of DSHS by three-dimensional limb mode. There are two locations of satellite at the time
       of T2 which meet the requirement of three-dimensional limb mode. Map data: © Google Earth, Image IBCAO and
       Landsat/Copernicus.

       So, the position of the time of T2-1 is picked up as the other satellite location where the field of view slices are orthogonal
to each other at the intersection region. The OH concentrations at different heights are obtained by the combinations of
       observed data from the SHS1 at the time of T1 and SHS2 at the time of T2 in the intersection region. Although the
       observed data are different because the geometric position and the viewing angle are of difference between the time of
       T1 and T2 the OH concentrations in the target area will also be constant. The data forms and the structures of the observed
       data of the SHS1 and SHS2 are the same in three-dimensional limb mode. The distance between the intersection point
R1 and tangent point P1 is presumed as 50 km which is used in the subsequent research.


# 3 Forward modeling

The forward model is used to express a physical process how the atmospheric state parameter is obtained by the sensors. A precise forward model is significant for the design of the instrument and the retrieval algorithm. The observed data received by the DSHS with an ultra-high spectrum resolution makes up by two parts in this research: the atmospheric

background radiance and the OH fluorescence emission radiance. These data can be simulated by a forward model which is constructed by the flow chart in the Fig. 6.

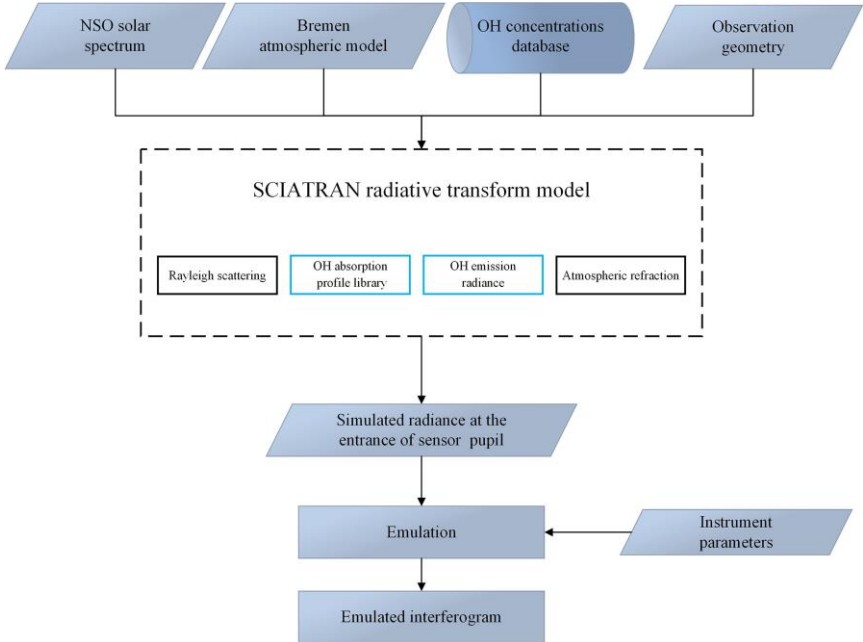

**Figure 6.** Flow chart of the Forward model.

The forward model is mainly constituted by a radiative transfer model called the SCIATARN which is developed for the Scanning Imaging Absorption spectrometer for Atmospheric CHartographY (SCIAMACHY). However, some parts of it need to modify for meeting the characters of the DSHS. The radiance source, the solar radiation, is considered in this

forward model first for the radiance transmission of OH in the middle and upper atmosphere because (1) OH fluorescence emission radiance will be generated from the excited state to the ground state when the OH is excited by the solar radiation. A high-precision calculation of the OH fluorescence emission radiance requires the high-resolution solar radiation, (2) the DSHS has a spectral resolution of 0.02 nm at least. The higher spectral resolution solar radiation will be used to get more realistic values. So, the National Solar Observatory (NSO) solar spectrum with resolution of $8.6\times10^{-4}$ nm is used for this forward model. In addition to the NSO, the precision of the OH spectrum is also significant.






SCIATRAN just takes the Rayleigh scattered radiance function, the ozone absorption function, the OH self-absorption function into account, but ignores the effect of OH fluorescence emission radiance. The Lifbase software is used to calculate the OH emission spectrum in a temperature ranges according to the Bremen atmospheric model. The OH emission spectrum database which put into the SCIATRAN as a source function is built up based on it. The corresponding

OH spectral emission data can get from the database when the observed radiance is simulated at some conditions. Next, the OH absorption spectrum database in the SCIATRAN bases on the HITRAN 2012 database which there are no OH absorption profiles in the ultraviolet band. The SCIATRAN absorption spectrum database is completed based on the HITRAN 2014 database which contains the OH absorption profiles to solve this problem. Finally, the precise atmospheric composition parameters are the assurance to simulate the observed radiance received by the sensors. That

means the high-quality OH concentrations database is needed when the radiance transfer is simulated accurately. For one thing, the MLS on the Aura has monitored the global OH for many years and the data can be acquired publicly at present. These data are used to build up the OH concentrations database. MLS was launched in15 July 2004 aboard on Aura, and kept offering consequent OH data until 2009. It was stopped monitoring from the November 2009 to the August 2011 in order to extend the service life of the sensor. It is restarted to monitor OH on the August and the

September each year continually to obtain annual change trends of OH data. For another thing, the solar activity was relatively stable and maintained at a low level in the whole solar activity cycle from 2005 to 2009. The solar activity from 2010 to 2016 had just ended an active cycle, and the level of solar activity in the next few years would be similar to the situation from 2005 to 2009. Therefore, the OH concentrations data from 2005 to 2009 is a great reference for the research of atmospheric OH concentrations by the DSHS in the next few years. The MLS OH concentrations Level 2

geophysical production is used and averaged within a certain range for reducing the random errors to build up the OH concentrations database. The N32 Gaussian grid from European Centre for Medium-range Weather Forecasts is selected as the averaged methods due to the random errors of MLS OH concentrations in high latitude are much larger than them in low latitude. It is a lattice-level coordinate system for scientific model of spheres in earth science. The latitude and longitude zones in the grid are divided unconventionally. The latitude band intervals of the northern and southern

hemispheres are symmetrical about the equator. The latitude band intervals and number of longitude bands on a latitude gradually decrease with the increase of the latitude to ensure that each grid area is approximately equal. In addition, the four seasons definition is used, the first quarter (March/April/May), the second quarter (June/July/August), the third quarter (September/October/November), the fourth quarter (December/January/February), as the temporal resolution for the OH concentrations database. A profile is given as an example in the (27 °N,106 °E) area in the Fig. 7. The season is

first quarter and the OH concentrations database is N32-seasonly database.



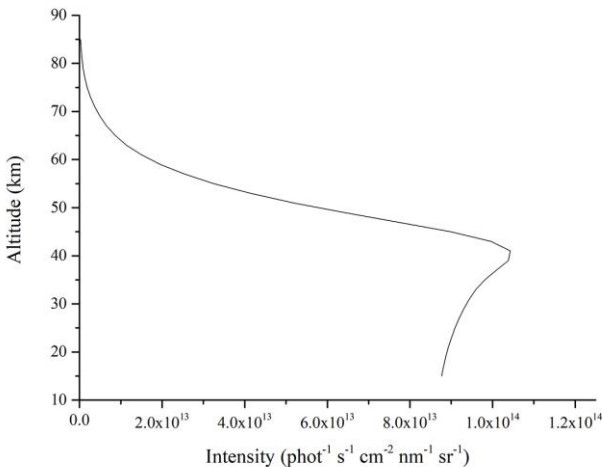

**Figure 7.** An observed radiance profile obtained by the modified SCIATRAN.

The results of modified SCIATRAN will be imaged in the image system which collects a complete data from all pixels on one line of CCD. The digital number (DN) values of each pixel in each row are generated by Eq. (4) at different tangent heights:

$$S_{CCD}(i,j) = \int_{y(j)}^{y(j+1)} \int_{x(i)}^{x(i+1)} \int_{0}^{\infty} B(\sigma, y) R(\sigma) \left(1 + \cos\{2\pi[4(\sigma - \sigma_0)x\tan\theta]\}\right) d\sigma dx dy \tag{4}$$


where $i$ is the number of rows along the line, $j$ is the tangent height from 15 to 85 km. The $x(i)$ and $y(j)$ is the areas where grating imaging image in the CCD, $B(\sigma, y)$ is the input radiance spectrum, $R(\sigma)$ is the instrument function, $\sigma$ is the wavelength number and the $\theta$ is the Littrow angle. The interferogram will be got based on the instrument parameters in the Table 1.


**Table 1.** Instrument parameters for emulation

| Parameters | value |
|---|---|
| Littrow length (nm) | 306 |
| Grating groove density (lines/nm) | 1000 |
| Numbers of interferogram samples | 1024 |
| Grating width | 15 |
| Littrow angle (°) | 8.80 |

An emulated interferogram is shown in the Fig. 8. It is the two-dimensional observed radiance interferogram constructing



by 1024 columns which indicate numbers of interferogram samples and 36 rows which indicate the observed tangent heights. The observed radiance gradually increases to reach the peak value which is the bright of interferogram with the altitude increases until the altitude around 40 km. The bright decreases gradually until the upper limit of the detection height.

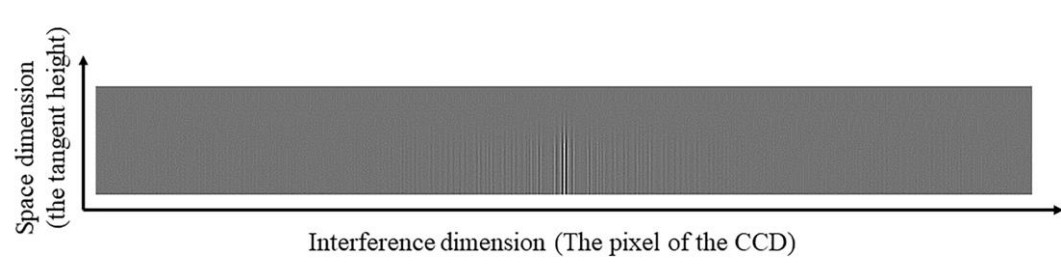


**Figure 8.** Emulated two-dimensional observed radiance interferogram.

The errors of this forward model mainly depend on three factors based on the modified parts above: the atmospheric model, the Doppler effect and the instrument calibration errors. First, the ozone in the atmospheric model is the influential

factor which is considered in the atmospheric model because its absorption cannot ignore in the ultraviolet band. Second, the satellite has different heliocentric speeds when it flies over the area where the local solar time is noon or is around the morning faint line. The speed will lead to the shift of observed radiance at the ranges of wavelength. Third, there are many sources of errors in the process of converting electrical signals into radiance values. The instrument calibration errors are assumed as $\pm 5\%$ because the DSHS has not been officially working. The total errors of observed radiance are

also given in the Table 2. Zhang had made the detailed analysis of errors and the reasons (Hong-hai et al., 2017).





**Table 2.** Total errors of observed radiance caused by different factors at some tangent heights


| Tangent height (km) | Errors caused by the atmospheric model (%) | Errors caused by Doppler effect (%) | Errors caused by instrument calibration errors (%) | Total errors (%) |
|---|---|---|---|---|
| 81 | From -2.65 to 5.67 | From -0.87 to 0.85 | From -5 to 5 | From -9.52 to 9.52 |
| 71 | From -2.61 to 5.62 | From -1.18 to 0.95 | From -5 to 5 | From -9.52 to 9.52 |
| 61 | From -2.81 to 5.83 | From -0.98 to 0.83 | From -5 to 5 | From -9.67 to 9.67 |
| 51 | From -4.90 to 8.17 | From -0.91 to 0.78 | From -5 to 5 | From -11.93 to 11.93 |
| 41 | From -16.49 to 26.17 | From -0.82 to 0.67 | From -5 to 5 | From -31.75 to 31.75 |
| 31 | From -18.92 to 36.64 | From -0.77 to 0.65 | From -5 to 5 | From -41.85 to 41.85 |
| 21 | From -19.56 to 39.10 | From -0.76 to 0.65 | From -5 to 5 | From -44.30 to 44.30 |

The DSHS is constituted by double spatial heterodyne spectrometers in the orthogonal layout. The distance between the intersection point and the tangent point is defined as 50 km and the angle between the spatial heterodyne spectrometer scanning direction and the vertical/horizontal direction is defined as 30°. The SHS2 positions at the time of T2 can be

calculated according to the SHS1 observation geometry parameters at the time of T1 in the target area based on the detection theory of three-dimensional limb mode. The simulated observation radiance of SHS2 and SHS1 can be obtained by the modified SCIATRAN above. The partial associated geometric parameters of the time of T1 and the calculated geometric parameters of the time of T2 are given in the following Table 3.

**Table 3.** Locations and geometry parameters of SHS1 at the time of T1 at some tangent height and the corresponding parameters of SHS2 at the time of T2

| Altitude (km) | The time of T1 | | | The time of T2 | | |
|---|---|---|---|---|---|---|
| | Latitude (°) | Longitude (°) | Azimuth angle (°) | Latitude (°) | Longitude (°) | Azimuth angle (°) |
| 21 | 54.501 | 121.483 | 321.98 | 55.071 | 121.534 | 321.041 |
| 31 | 54.501 | 121.483 | 321.98 | 55.117 | 121.303 | 321.045 |
| 41 | 54.501 | 121.483 | 321.98 | 55.163 | 121.073 | 321.050 |
| 51 | 54.501 | 121.483 | 321.98 | 55.210 | 120.844 | 321.056 |
| 61 | 54.501 | 121.483 | 321.98 | 55.255 | 120.616 | 321.062 |
| 71 | 54.501 | 121.483 | 321.98 | 55.31 | 120.389 | 321.070 |
| 81 | 54.501 | 121.483 | 321.98 | 55.346 | 120.163 | 321.078 |

The intersection region of the time of T1 and T2 mentioned above is (67.46 °N,86.618 °E). The observed radiance





received by SHS1 at the time of T1 and received by SHS2 at the time of T2 is composed by a group of different satellite
positions at tangent heights which shown in the Fig. 9.

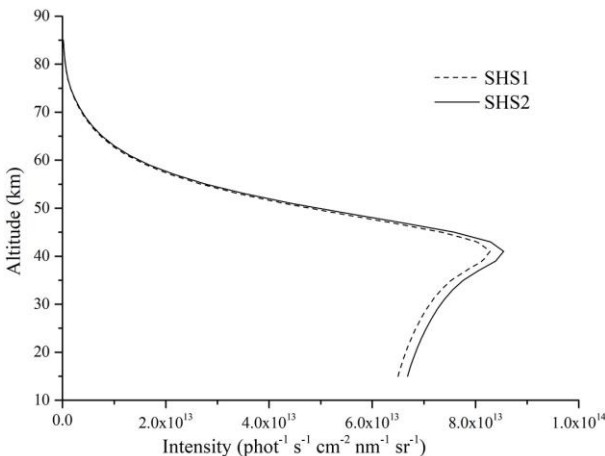

**Figure 9.** The simulated observation data in the intersection region at the time of T1 and T2. The dash line is the simulated
observation data received by SHS1 at the time of T1 and the solid line is the simulated observation data received by the
SHS2 at the time of T2.

Although the observation geometries and satellite positions are different at different time, both spatial heterodyne
spectrometers scan the atmosphere by the same limb mode, the observed radiance profile still maintains the distribution
in the range from 308 to 310 nm band. There are four global observed radiance databases for this research. The first
quarter observed radiance is given as an example: the NSO solar spectrum is used as the radiance source ,the Bremen
global atmospheric model, the first quarter OH concentrations data of N32-seasonly OH concentrations database and the
geometric parameters of MLS OH production is also used to simulate the observed radiance in the range from 308 to
310 nm band at the time of T1 and T2. The results are shown in the Fig. 10 which the observed radiance received by
SHS1 at the time of T1 and Fig. 11 which the observed radiance received by SHS2 at the time of T2.





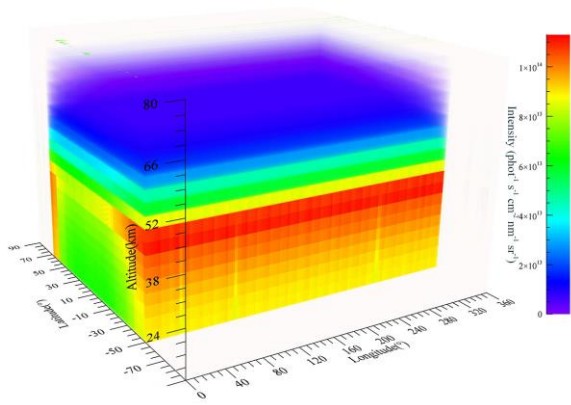

**Figure 10.** The simulated observation radiance received by SHS1 at the time of T1 under the different conditions

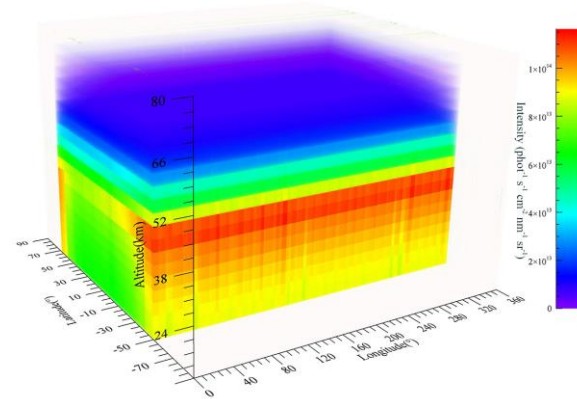


**Figure 11.** The simulated observation radiance received by SHS2 at the time of T2 under the different conditions

There are some differences between the Fig. 10 and the Fig. 11 because of the observed geometries and satellite positions.

However, the whole observed radiance trends are same. The observed radiance increases from $8\times10^{13}$ phot$^{-1}$ s$^{-1}$ cm$^{-2}$ nm$^{-1}$ sr$^{-1}$ to $1.4\times10^{14}$ phot$^{-1}$ s$^{-1}$ cm$^{-2}$ nm$^{-1}$ sr$^{-1}$ until the altitude around 45 km and decreases to about $4\times10^{13}$ phot$^{-1}$ s$^{-1}$ cm$^{-2}$


nm$^{-1}$ sr$^{-1}$ with the altitudes increase. The lowest observed radiance is in the region around the 30 °N and gradually
increases towards to the two poles in the altitude of 41 km. The reason for this trend is that the solar zenith angle is the
minimum value around the 30 °N, and then increases towards to the two poles. At the altitude of 71 km, the observed
geometries are not the only influencing factor and the distributions of OH concentrations are another one. The Rayleigh
scattered radiance is weak and OH fluorescence emission radiance becomes strong. In addition to this one, the OH
concentrations at the first quarter are low in the region around 60 °N that lead to the high radiance values appear in the
southern hemisphere.

# 4 Inversion modeling

## 4.1 Interferogram pretreatments


The method of forming the interferogram by the spatial heterodyne spectroscopy is the spatial heterodyne modulation.
The signal-to-noise ratio is susceptible to many factors, and the fine spectral lines of noisy signal can also be detected
simultaneously. The interferogram needs to be pretreated for solving these problems and transformed to obtain the OH
fluorescence emission radiance. The whole pretreatments are shown in the Fig. 12.


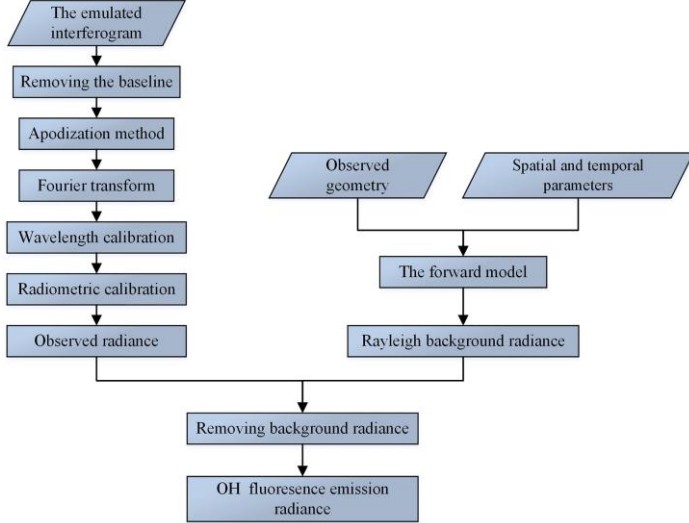

**Figure 12.** Flow chart of the interferogram pretreatments.

First, the interference data with low frequency baseline will make low frequency spurious signals in the process of the
Fourier spectrum transform. The methods of removing the baseline mainly include the polynomial linear fitting to
baseline, the first-order differential de-baseline at present. The function of first-order differential de-baseline method is





the high-pass filtering. It is the most useful way to remove the baseline (Song et al., 2009). The interferogram obtained

by the spatial heterodyne spectrometer is the interference data in the interval of finite optical path difference from -L to +L. That means an interferometric function forces a sudden drop to zero outside of this range. It will cause the interferogram to appear sharp discontinuities in the edge interval. The spectral profile has side lobes which the positive side lobes are the source of the false signal, and the negative side lobes will lead to the adjacent weak spectral signals to be submerged. The apodization method is used to mitigate the discontinuity of interferogram edge through multiplying the interferogram function by a progressive weight function. The interferogram after apodization is subjected to the

Fourier spectrum transform for obtaining the corresponding spectrum. However, the high-spectral resolution observed radiance data which is simulated by the forward model mentioned above includes the atmospheric background radiance which must be identified and removed from the observed radiance data before the OH fluorescence emission radiance is obtained because: (1) it accounts for more than 95% of observed radiance that will make OH fluorescence emission radiance low, (2) some errors of atmospheric background radiance will transfer into the OH fluorescence emission

radiance when the observed radiance of the interferogram form is recovered to the spectral form. The OH fluorescence emission radiance graph which is shown in the Fig. 13 can obtain after these steps and the Fourier spectrum transform can be done.

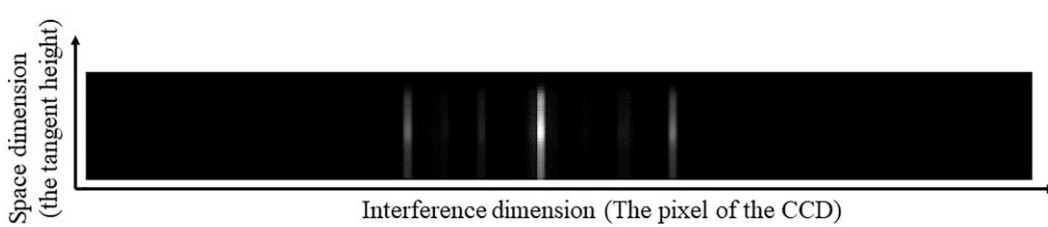


**Figure 13.** Graph of OH fluorescence emission radiance.

The fifth step is the wavelength calibration. The zero position of the spectral data points corresponds to the Littrow wavelength number in the Fourier transform spectrum, so the wavelength calibration equation is given by the Eq. (5):


$$\delta_i = -1.3166574i + 32679.738 \tag{5}$$

where $i$ is from 0 to 1024. The sixth step is the radiometric calibration. The purpose of the radiometric calibration is to determine the quantitative relationship between the output signal of instrument and the spectral radiance because the

spectrometer obtains the DN values directly. A liner fit method is used to simulate the different sets of known radiance values at each spectral data point for establishing a fitting relationship in the calibration process by Eq. (6):

$$S_\delta = B_\delta \cdot K_\delta + \varepsilon_\delta \tag{6}$$


where $S_\delta$ is the DN values at the wavelength number $\delta$. $B_\delta$ is the assumed incident spectrum radiance. $K_\delta$ is the
calibration factor and $\varepsilon_\delta$ is the deviation caused by other factors. A spectrum of OH fluorescence emission radiance is
shown in the Fig. 14.

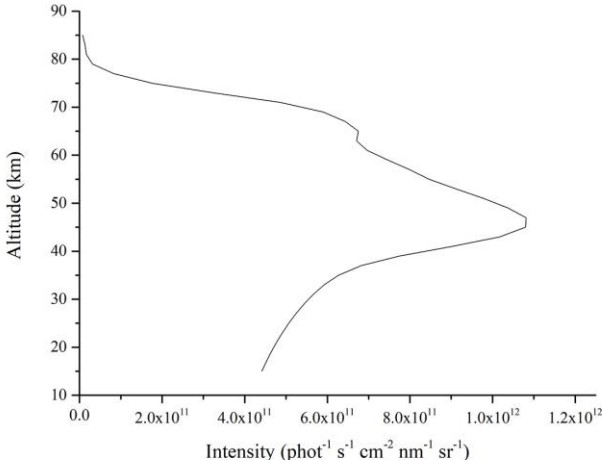

**Figure 14.** OH fluorescence emission radiance spectrum extracted from observed radiance.

## 4.2 Constructions of the retrieval algorithm

It is an inversion process to obtain the OH concentrations by the DSHS observed radiance. The observed radiance is
simulated through the forward model in this research. However, it cannot be done to obtain OH concentrations in a
converse way directly. The atmospheric inversion problems are not an inversion of the forward model but use some
related algorithms to estimate the atmospheric parameters for finding the best state parameters with the observed radiance
data.

The DSHS consists by two spatial heterodyne spectrometers in the orthogonal layout for monitoring the OH from 15 to
85 km. It will provide large and complicated data. The traditional iterative retrieval algorithm will take more time to
obtain the OH concentrations in this case because of the character of iteration. A new retrieval algorithm which is suitable
for the DSHS needs to be constructed for inverting the accurate OH concentrations.

The precondition of the three-dimensional limb mode for monitoring the OH is the time interval must be relatively short,
otherwise the OH concentrations will change, and the observed data cannot reflect the accurate OH concentrations. The
distance between the satellite positions at the time of T1 and T2 is approximately 66 km when the intersection point is
assumed as 50 km away from the tangent point mentioned in the Sect. 2. So, the intervals between those points are very
small. Additionally, the OH concentrations do not change much in a very short time according to MLS products. Short


interval time between the time of T1 and T2 and small change of OH concentrations in a very short time support the theoretical premise of the new retrieval algorithm and make DSHS possible to detect atmosphere through the three-dimensional limb mode. An algorithm called tomographic retrieval algorithm is constructed to invert OH concentrations from 15 to 85 km based on the theory above. The core of this retrieval algorithm is look up table method. The most

important step for it is the construction of the tomographic observed database. The main influencing factors for making up the tomographic observed database contain the atmospheric model, the observation geometries (the solar zenith angle and the relative azimuth angle), the spatial information (position of the target area and satellite position) and date parameters, the OH concentrations and the simulated observation data. The solar zenith angle and the relative azimuth angle have a greater influence on the observed radiance. The spatial information and date parameters will influence the

OH concentrations and the atmospheric model. The solar zenith angle is changed between 0° to 100° and the relative azimuth angle is changed between 0° to 180°. The corresponding OH concentrations are found out according to the date parameters and spatial information such as the season, latitude and longitude of the target area from the OH concentrations database which has been built up in the Sect.3. The OH concentrations are changed in the ranges between 50% to 150% times amount of the original corresponding one at different tangent height for more possible situations.

Many combinations of various state parameters can be generated according to these changes. The observed data received by DSHS in these combinations is simulated through the forward model mentioned above. The tomographic observed database can be built up for the tomographic retrieval algorithm. The flow chart is shown in the Fig. 15 and a subset of rows of the tomographic observed database is given in the Table 4 which the SZA is the solar zenith angle, the AA is the relative azimuth angle, the MultiND_height is the tangent height.


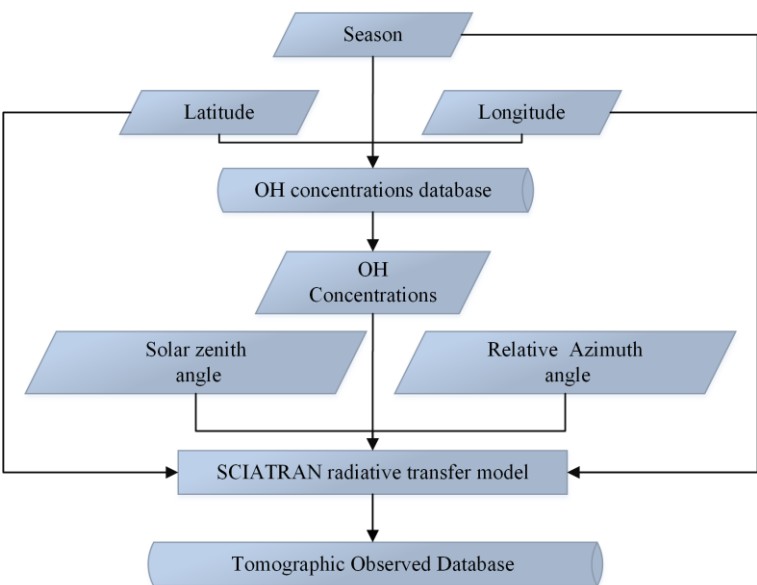

**Figure 15.** Construction of the tomographic observed database.



**Table 4.** A subset of rows of the tomographic observed database

| N | Latitude (°) | Longitude (°) | SZA (°) | AA (°) | MultiND_height (km) | Season |
|---|---|---|---|---|---|---|
| 1 | 54.240 | 125 | 0 | 0 | 15 | 1 |
| 2 | 54.240 | 125 | 0 | 0 | 15 | 2 |
| 3 | 54.240 | 125 | 0 | 0 | 15 | 3 |
| 4 | 54.240 | 125 | 0 | 0 | 15 | 4 |
| 5 | 54.240 | 125 | 0 | 0 | 17 | 1 |
| 6 | 54.240 | 125 | 0 | 0 | 17 | 2 |
| 7 | 54.240 | 125 | 0 | 0 | 17 | 3 |

Based on the satellite positions, the observation geometries, the date parameters, the observed radiance of SHS1 at the time of T1 and the corresponding observed radiance of SHS2 at the time of T2, the OH concentrations in the target area can be found in the tomographic observed database directly. If there is no corresponding query condition in the database, the cubic spline interpolation is used to calculate the OH concentrations. The cubic spline interpolation not only has the higher stability than other interpolation algorithms, but also can maintain the continuity and smoothness of the interpolation function under the premise of convergence. Figure 16 shows the flow chart that the OH concentrations obtain by the tomographic retrieval algorithm.

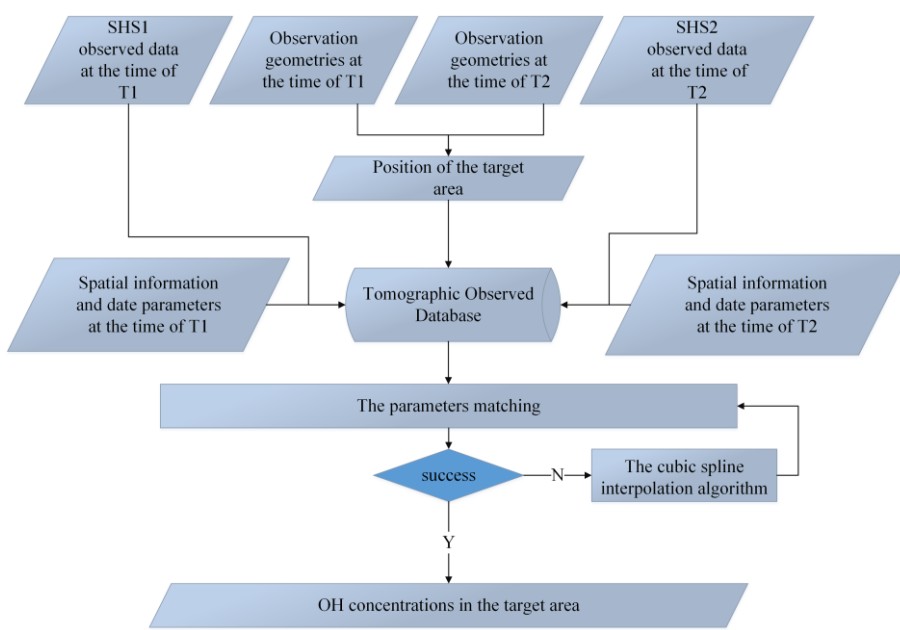

**Figure 16.** Flow chart of the tomographic retrieval algorithm. The OH concentrations can be obtained from the tomographic observed database or by the cubic spline interpolation algorithm.



# 5 Results and discussion

## 5.1 Inversion results

There are four tomographic observed databases based on the four seasons, here the spring season tomographic observed database is used as an example to prove the feasibility and superiority of the tomographic retrieval algorithm. The OH concentrations are obtained in the (67.46 °N, 86.168 °E) through satellite positions, observation geometries, date parameters which are given by the Table 4. If there is a corresponding query condition, the OH concentrations can be given directly from the tomographic observed database, otherwise the OH concentrations will be calculated by the cubic
spline interpolation. The inversion results are shown in the Fig. 17.

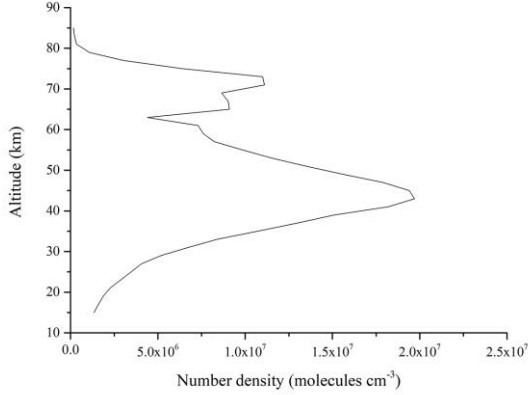

**Figure 17.** Inversion results in the target area using the tomographic retrieval algorithm.

The results indicate that a maximum peak value of OH concentrations are about $2.0 \times 10^7$ molecules cm$^{-3}$ appear around the 45 km height as the heights increase which closely relates to the ozone concentrations in the stratopause. A valley value of OH concentrations about $5.0 \times 10^6$ molecules cm$^{-3}$ appear around 65 km. As the height continues to rise, the second maximum peak value of OH concentrations about $1.0 \times 10^7$ molecules cm$^{-3}$ appears around the 70 km height which
is mainly affected by the water vapor concentrations in the mesosphere. The OH concentrations above the 75 km height and below the 25 km is very low (about $2.5 \times 10^6$ molecules cm$^{-3}$). In general, the OH concentrations increase with the altitudes rise until around 40 km height and reach the valley value around 65 km. The second maximum value peak which is affected by the water vapor concentrations appears around 75 km, then the OH concentrations continue to decrease until the limited altitudes of detection.



## 5.2 Discussion


We also constructed an iterative retrieval algorithm to compare with the tomographic retrieval algorithm here for reflecting the outstanding advantages of tomographic retrieval algorithm. The schematic diagram of iterative inversion is shown in the Fig. 18. The simulated radiance can be obtained under the condition which an initial OH concentrations guess can get from the OH concentrations database. The residuals of the simulated and observed data will be calculated

to judge the convergence. The initial OH concentrations guess values are the inversion results if the residual meets the accurate requirements of research. Otherwise, the retrieval algorithm will be used to obtain the inversion results.

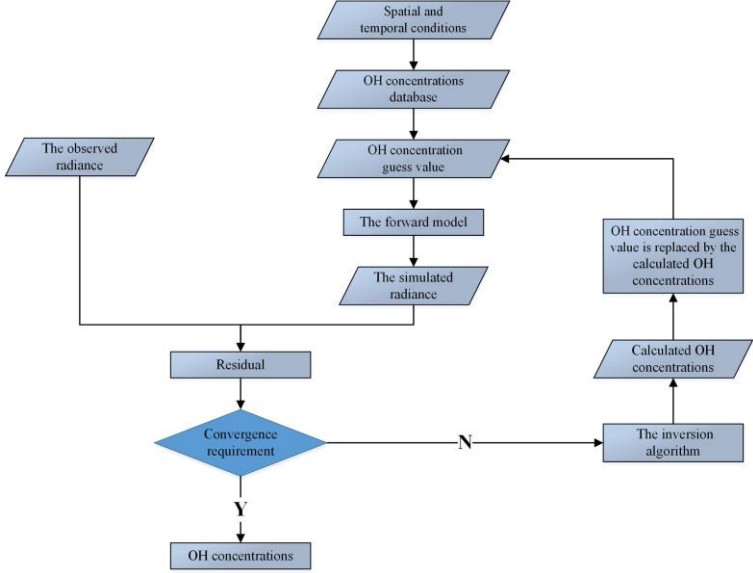

**Figure 18.** Schematic diagram of iterative inversion.

The LSUV (Limb scan of Scattered UV radiation) retrieval algorithm (Aruga and Heath, 1988;Aruga and Igarashi, 1976) is modified to invert OH concentrations at each tangent height. The tangent height is defined as $m$ and the target area is divided into $n$ atmospheric layers from the bottom to the top. One-order approximate Taylor expansion is used to establish

a linear relationship between the OH concentrations and observed radiance at each atmospheric layer by Eq. (7):

$$F_i = \left[\sum_{j=1}^n D_{ij} \cdot y_j\right]\varepsilon_i \quad \begin{cases} i = 1, \dots, n \\ j = 1, \dots m \end{cases} \tag{7}$$

where $\varepsilon_i$ ,the inversion coefficient, is approximately 1 when the OH concentrations are closed to the guess OH

concentrations. The $F_i$ is the difference between the observed data and simulated data by Eq. (8) at the $i$ layer:





$$F_i = I(h_i) - I_s(h_i), i = 1,2,\ldots,m \tag{8}$$

The $D_{ij}$ which is given by Eq. (9) is the partial derivative of radiance data at the $i$ tangent height:


$$D_{ij} = \left[\frac{\partial F_i(y_j)}{\partial y_j}\right]_{y_j=0} = \frac{I_s{}^*\left(h_i, \frac{\Delta y_j}{2}\right) - I_s{}^*\left(h_i, -\frac{\Delta y_j}{2}\right)}{\Delta y_j} \tag{9}$$

The $y_j$ in the Eq. (10) is relative error between the OH concentrations $N(j)$ and the guess concentrations $N_g(j)$:

$$y_j = \frac{[N(j) - N_g(j)]}{N_g(j)} \quad j = 1,2,\ldots,n \tag{10}$$

The approximate OH concentrations at each tangent height $(Q)_i$ can be got based on the Eq. (10) through the Eq. (11):

$$(Q)_i = \frac{F_i}{\sum_{j=1}^{n} D_{ij}} \tag{11}$$


The approximate OH concentrations $(Q)_i$ at $i$ tangent height is determined by the OH concentrations on several atmospheres near the $i$. The weight function $P_{ij}$ is established from the $D_{ij}$ by Eq. (12):

$$P_{ij} = \frac{D_{ij}}{\sum_{j=1}^{n} D_{ij}} \tag{12}$$


where $P_{ij}$ is the proportion of the contribution of OH concentrations on the radiance of each tangent height in each layer. The accurate relative errors of the OH concentrations and guess OH concentrations in each layer $y_j$ can be obtained from $P_{ij}$ by Eq. (13):

$$y_j = \frac{\sum_{i=1}^{m} Q_j \cdot P_{ij}}{\sum_{i=1}^{m} \cdot P_{ij}} \tag{13}$$

However, the difference between the OH concentrations and guess OH concentrations is large in the actual satellite detection, the $\varepsilon_i$ is not 1, the $\varepsilon_i{}^*$ is used to replace it by Eq. (14):

$$\varepsilon_i{}^* = \frac{F_i(y_1{}^*, y_2{}^*, \ldots, y_n{}^*)}{\sum_{j=1}^{n} D_{ij} y_j{}^*} \tag{14}$$



Finally, the precise relative error $y_j$ between the OH concentrations and the guess OH concentrations can be obtained. The OH concentrations $N(j)$ can get though the $y_j$ and $N_g(j)$. The convergence error $R_i^l$ between the observed data $I(h_i)$ and simulated data $I_s(h_i)$ at $i$ layer which based on the guess OH concentrations $N_g(j)$ is calculated by Eq. (15):

$$R_i^l = \frac{I(h_i) - I_s^{(l)}(h_i)}{I_s^{(l)}(h_i)} = \frac{F_i^{(l)}}{I_s^{(l)}(h_i)} \tag{15}$$

where $l$ is the number of iterations. The average residual $R^l$ based on the $R_i^l$ at different tangent heights can be calculated by Eq. (16):

$$R^l = \sqrt{\frac{\sum_{i=1}^{m} \{R_i^l\}^2}{m}} \tag{16}$$

The threshold of average residual and the numbers of iteration are set based on the inversion experience. The iteration result which meets the precision requirement is the final result, otherwise the prior $N_g(j)$ will be replaced by the $N(j)$ which is calculated by the inversion algorithm and the iteration process will continue until the residual converges meet the accurate requirement or the number of iterations exceeds the iteration number threshold. The iteration stops when the average residual decreases below 0.005 or the number of iterations reaches 15 times in this research.

The observed radiance which is simulated by the forward model are used by the LSUV retrieval algorithm to invert OH concentrations. When the iteration results meet the accurate requirements of research, the OH concentrations can be obtained. The OH concentrations obtained by the LSUV retrieval algorithm are credible in the upper atmosphere. These results are the same as the results of MAHSRI and SHIMMER (Conway et al., 1999;Englert et al., 2010). However, the OH concentrations in the lower atmosphere such as below 30 km are unsuitable for scientific research. That is the reason why there are no OH concentrations in these regions from the MAHRSI and SHIMMER (Harlander et al., 2002). However, the tomographic retrieval algorithm effectively avoids the constraints of the initial guess values and does not take a lot of time to iterate for obtaining the results in the inversion progress. The errors of this retrieval algorithm just include the errors caused by the interpolation method and the observed radiance errors which are used in this algorithm. The observed radiance errors are affected by the atmospheric model, the Doppler effect and the instrument calibration errors. The OH fluorescence emission radiance which is separated from the observed radiance is the most important part for the OH concentrations, so the contributions of each part to the OH fluorescence emission radiance is analyzed by the following method: the undisturbed parts are taken as the actual parameters, and the results based on these parts are the true OH fluorescence emission radiance. Then, some certain amounts as the disturbances are applied to the three parts above. The OH fluorescence emission radiance based on these parts is the incorrect results. The relative errors between the true and incorrect OH fluorescence emission radiance is calculated by the Eq. (17):





$$Err = \frac{[I]_{err} - [I]_{act}}{[I]_{act}} \times 100\%$$ (17)

where the $[I]_{err}$ is the incorrect OH fluorescence emission radiance and the $[I]_{act}$ is the true OH fluorescence emission radiance.

## 5.2.1 Influence of the atmospheric model

The atmosphere environment is complicated. Although, the atmospheric model used in this research is scientific and reasonable, it cannot reflect the accurate atmospheric condition which leads to some errors between true OH fluorescence emission radiance and incorrect OH fluorescence emission radiance. These will cause an error when the OH concentrations are looked up or calculated. Many factors can affect the OH fluorescence emission radiance in the atmospheric model such as some physical parameters like temperature, pressure and some atmospheric composition like

nitrogen dioxide and formaldehyde. Here, the main composition we consider is ozone which is the source of OH and mainly affects the radiance transmission in the ultraviolet band. The original ozone profiles are assumed as the actual atmospheric state parameter and the amount of ozone profile for the disturbances is set as ±30%. The relative errors of the OH fluorescence emission radiance caused by the atmospheric model is calculated by Eq. (17) and are shown in the Fig. 19. The relative errors of the OH fluorescence emission radiance at some tangent height are also given in the Table

600   5.

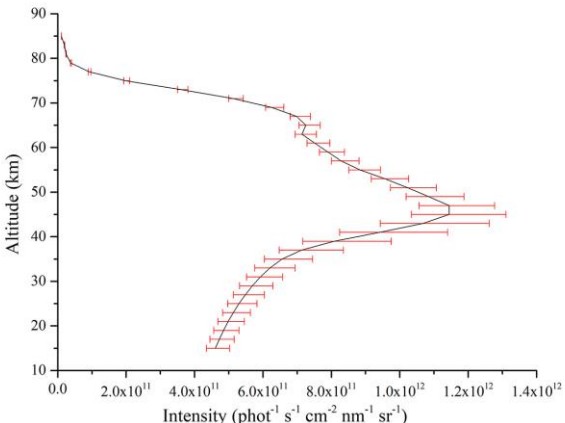

**Figure 19.** Relative errors of OH fluorescence emission radiance caused by the atmospheric model in the tomographic retrieval algorithm. The red error bars indicate the relative errors caused by the atmospheric model.




**Table 5.** Relative errors of OH fluorescence emission radiance caused by the atmospheric model at some tangent heights

| Tangent height (km) | Relative errors caused by the atmospheric model (%) |
|:---:|:---:|
| 81 | From -2.71 to 5.72 |
| 71 | From -2.72 to 5.75 |
| 61 | From -2.96 to 5.92 |
| 51 | From -4.92 to 8.19 |
| 41 | From -12.57 to 20.92 |
| 31 | From -6.78 to 10.98 |
| 21 | From -6.00 to 9.51 |


The relative errors caused by the atmospheric model indicate that the atmospheric model have a strong impact on the OH fluorescence emission radiance especially in the lower atmosphere where the ozone concentrations are high. The errors caused by ±30% uncertainty of the atmospheric model have a peak value from -12.57 % to 20.92 % at the altitude of 41 km. It also appears in the LSUV retrieval algorithm by the same calculation method which is from -12.19 % to

20.28 %. The concentrations of ozone gradually decrease with the relative errors decreasing to about -3% to 5% when the heights increase. The effect the of atmospheric model is smaller in the upper atmosphere. The distribution of errors has the same tendency in the LSUV retrieval algorithm that the relative errors increase and then gradually decrease after reaching the peak value altitude. These mean the relative errors caused by the atmospheric model are inevitable. If the more actual atmospheric model is used, the more accurate OH fluorescence emission radiance may be obtained which is

the key for the retrieval algorithm.

## 5.2.2 Influence of the Doppler effect

The satellite has the different heliocentric speeds on its working orbit around the earth at different times. It will cause a pseudo-random offset of observed data within $\pm 5 \times 10^{-4}$ nm at the range of wavelength. The observed radiance is calculated within $\pm 5 \times 10^{-4}$ nm disturbance at the wavelength. The OH fluorescence emission radiance which is separated

from the observed radiance is too weak.

A little Doppler effect will cause a great influence on the weak OH fluorescence emission radiance. The magnitude of wavelength shift caused by the Doppler effect is much smaller than the spectral resolution. "Finding peak" is used to find a peak value which needs to be larger than the values on its sides. It is impossible to correct the offset by "Finding Peak" method. However, "Taking peak" is defined to find the highest peak value in the OH fluorescence emission

radiance. The "Taking Peak" is used for reducing the impact of the Doppler effect. Therefore, the OH fluorescence





emission radiance at the peak position is used instead of the wavelength from 308 nm to 310 nm when calculating the relative errors caused by the Doppler effect at a single tangent height. The relative errors of the OH fluorescence emission radiance caused by the Doppler effect is calculated by Eq. (17) and are shown in the Fig. 20. The relative errors of OH fluorescence emission radiance caused by the Doppler effect at some tangent heights are also given in the Table 6.


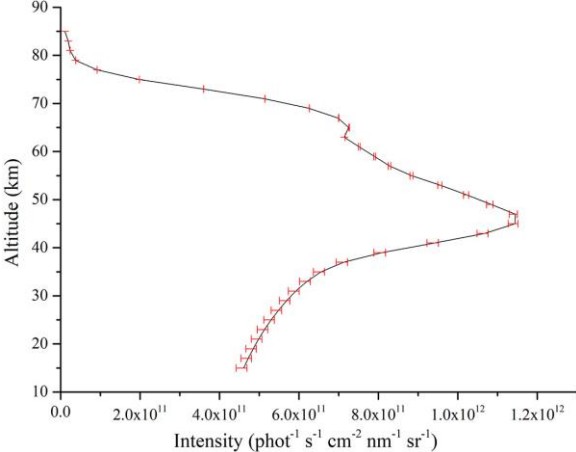

**Figure 20.** Relative errors of OH fluorescence emission radiance caused by the Doppler effect in tomographic retrieval algorithm. The red error bars indicate the relative errors caused by the Doppler effect.


**Table 6.** Relative errors of OH fluorescence emission radiance caused by the Doppler effect at some tangent heights

| Tangent height (km) | Relative errors caused by the Doppler effect (%) |
|---|---|
| 81 | From -0.39 to 0.35 |
| 71 | From -0.20 to 0.31 |
| 61 | From -0.11 to 0.34 |
| 51 | From -0.88 to 0.41 |
| 41 | From -2.16 to 0.93 |
| 31 | From -3.14 to 1.44 |
| 21 | From -3.71 to 1.62 |


The relative errors of OH fluorescence emission radiance caused by the Doppler effect decrease when the altitudes rise. The distributions of errors have the same tendency, but the relative errors of tomographic retrieval algorithm are smaller than LSUV retrieval algorithm. These indicate that tomographic retrieval algorithm can obtain more accurate OH concentrations although the Doppler effect cannot avoid.

## 5.2.3 Influence of other factors

There are some other errors exist apart from the errors above. For one thing, there are various sources make some errors between the radiance received by the sensors and the actual radiance in the instrument calibration process. The instrument calibration errors will cause an error of the inversion result. We assumed the instrument calibration errors are $\pm5\%$ in this research after referring to the detector parameters due to the DSHS do not officially work. For another, the error caused by the interpolation method is the main factor we need to calculate. The OH concentrations can be obtained by
the tomographic retrieval algorithm according to the atmospheric model, spatial information, date parameters, observation geometries and observed radiance. The interpolation errors are also the other error factors which effect the accuracy of OH concentrations. It is about 0.32% for the six-dimensional cubic spline interpolation of OH concentrations, date parameters, longitude, latitude, solar zenith angle and azimuth angle after research and related experiment. Therefore, 0.32% is taken as the error of the interpolation algorithm in the tomographic retrieval algorithm.

## 5.2.4 Analysis of total errors


The theoretical basis for calculating the total errors of inversion results is the error transfer formula which is always used to calculate the indirect measurement error (Hong et al., 2009).The total errors based on a single error factor often have an asymmetric distribution. The corresponding standard error according to the single each error factor should be calculated if error transfer formula is used. It is calculated according to the B-class standard uncertainty evaluation
method. The relationship between input parameters and results can be expressed by the Eq. (18):

$$[I] = f(x_1, x_2, x_3, \ldots, x_n) \tag{18}$$

where $x_1, x_2, x_3, \ldots, x_n$ are the input parameters and $[I]$ indicates the inversion results. Equation (19) can be got when
the error $\Delta x_i$ is considered and the does a Taylor expansion:

$$[I] + \Delta[I] = f(x_1, x_2, x_3, \ldots, x_n) + \sum_{i=1}^{n} \frac{\partial f}{\partial x_i} \Delta x_i \tag{19}$$





The maximum value of relative error can be calculated by the Eq. (20):

$$\frac{\Delta[I]}{[I]} = \sum_{i=1}^{n} \left| \frac{\partial f}{\partial x_i} \right| \frac{\Delta x_i}{[I]} \tag{20}$$

where the right part of equal sign means the errors caused by $x_1, x_2, x_3, \ldots, x_n$. The error transfer formula can be derived from the Eq. (20) when the errors are squared. The total errors RSS can be calculated by the "square root" method with the Eq. (21):

$$RSS = \sqrt{\sum_{i=1}^{n} \left\{ \frac{\partial[I]}{x_i} \frac{\Delta x_i}{[I]} \right\}^2} \tag{21}$$

The total errors of the results caused by each factor can be obtained by the Eq. (21). The total errors of the inversion results caused by each error factor are calculated which are shown in the Fig. 21 and the total errors of inversion results at some tangent heights are given in the Table 7.

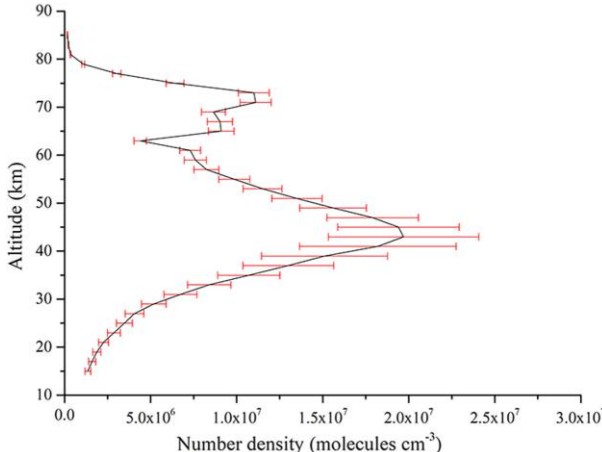

**Figure 21.** Total errors of inversion results caused by some factors mentioned above. The red error bars indicate the relative errors caused by some factors mentioned above.



**Table 7.** Total errors of inversion results at some tangent heights

| Tangent height (km) | Total errors of inversion results (%) |
|---|---|
| 81 | From -8.09 to 8.09 |
| 71 | From -8.11 to 8.11 |
| 61 | From -8.31 to 8.31 |
| 51 | From -10.83 to 10.83 |
| 41 | From -25.03 to 25.03 |
| 31 | From -14.27 to 14.27 |
| 21 | From -12.96 to 12.96 |


As the Table 7 indicates that the total errors of OH concentrations are increasing as the heights rise. It reaches the maximum value range from -25.03 % to 25.03 % at the 41 km height, and then decreases from -8.09 % to 8.09 % as the heights continue to rise until the limited height. The total errors of the inversion results in the lower atmosphere are small especially. It is a great improvement to compare with the results of the LSUV retrieval algorithm which has a

characteristic of iteration. The initial guess values have a great influence on the iterative speed and the inversion results for the iterative retrieval algorithm. A realistic initial guess value can achieve higher iteration efficiency and more accurate inversion results. The initial guess values used in the LSUV retrieval algorithm are obtained from the OH concentrations database through the date parameters and spatial information such as the season, latitude and longitude of the target area. The corresponding accuracy ranges of MLS OH concentrations products are also given, and its useful

height ranges are from 23 to 81 km. Therefore, the data accuracy is assigned at the upper limited height to altitude greater than the upper limit of MLS data altitude and at the lower limited height to the altitude smaller than the lower limit of MLS data altitude. The initial guess values can be obtained by using the data accuracy to perform the positive and negative turbulence for the initial OH concentrations values. A minimum positive value is given to the data less than zero for the physical meaning when a negative value may occur in the negative direction. The two initial guess values

are used in the LSUV retrieval algorithm separately. Although, the initial guess values above are chosen scientifically and reasonably, the inversion results are unsuitable for scientific research in the lower atmosphere that the relative errors caused by the initial guess values reach from -95.39 % to 545.44 %. Besides the initial guess values, the inversion results are also affected through the errors caused by the atmospheric model, the Doppler effect and the instrument calibration error. The relative errors of inversion results obtained by the LSUV retrieval algorithm are small from 30 km

to 85 km. The total errors of inversion results are from -24.83 % to 24.83 % in the altitude of 31 km and then decrease gradually until the altitude of 51 km. A little increase occurs from the altitude of 60 km to 70 km and reaches from -37.01 % to 37.01 % at the 81 km. However, the inversion results are bad in the lower atmosphere. The total errors of inversion results are from-553.16 % to 553.16 % in the altitude of 21 km. The total errors of inversion results are 2 times larger than tomographic retrieval algorithm at some heights like 71 and 61 km, however, the total errors of inversion

results are 4.625 times larger than the errors of the tomographic retrieval algorithm at 21 km height especially. The



application altitudes of the LSUV retrieval algorithm are limited. The main reasons lead to these unscientific inversion results are two parts. The OH fluorescence emission radiance, the main factor for the retrieval algorithm, is not strong enough due to the ozone optical depth becomes large in the lower atmosphere. That will lead to the observed radiance does not come from the tangent area but the near field. The lower atmospheric environment is complicated. The human activity will produce a lot of substances which will join in the atmospheric environment by the atmospheric convection motion. The aerosol also distributes here. These will lead to a bad detection accuracy. So, the useful altitudes of the LSUV retrieval algorithm based on the data of single spatial heterodyne spectrometer are limited. The DSHS can counteract the errors by the two spatial heterodyne spectrometers because of its special optical design which the inversion results of retrieval algorithm have proved. These analyses above also indicate that the tomographic retrieval algorithm is a feasible retrieval algorithm for the atmosphere from 15 to 85 km. The look up table method avoids the intervention of initial guess values effectively by establishing a tomographic observed database of multi-dimensional variables and is understood easily. It also avoids the complicated iterative optimization process, and the OH concentrations can be obtained directly from the tomographic observed database. These make the speed of inversion process fast compared with the LSUV retrieval algorithm. The tomographic retrieval algorithm is a suitable retrieval algorithm for the three-dimensional limb mode of DSHS as well. The three-dimensional limb mode provides big and numerous observed data. The speed of traditional iterative retrieval algorithms is slow and will cost a lot of time. The tomographic retrieval algorithm solves these problems and improves the efficiency of retrieval algorithm which is important for the OH concentrations. The LSUV retrieval algorithm can invert OH concentrations well under the conditions that the OH fluorescence emission radiance is strong and the interference factors like water vapor and ozone are little. However, a single spatial heterodyne spectrometer with traditional limb mode cannot obtain enough and high-quality data to invert the OH concentrations that will make the inversion results unscientific especially in the lower atmosphere. The initial guess values play a significant role in the LSUV retrieval algorithm due to the feature of the iteration. The determination of the initial guess values involves many factors and there is no best way to determine the initial guess values at present. This feature is particularly pronounced at the lower atmosphere where OH fluorescence emission radiance is less sensitive to OH concentrations.

In summary, the LSUV retrieval algorithm based on the iterative method can invert the OH concentrations well in the higher atmosphere like mesosphere but cannot obtain the accurate OH concentrations in the lower atmosphere like the bottom of stratosphere. Many factors lead to this result like the atmospheric model, the limit of initial guess values and something else. The tomographic retrieval algorithm using the look up table method can obtain the accurate OH concentrations from stratosphere to the mesopause with the feature of the three-dimensional limb mode. This retrieval algorithm not only saves the time of inversion process, but also avoids the problem that the OH concentrations are unsuitable for scientific research in the lower atmosphere.

# 6 Conclusions

The OH is the key oxidant in the atmosphere and has a great influence on the atmospheric photochemistry process. The





DSHS based on the spatial heterodyne spectroscopy will monitor the OH with the three-dimensional limb mode in the future. A forward model is constructed to simulate the observed data of DSHS accurately. A new retrieval algorithm for obtaining the OH concentrations is also proposed based on the simulations of the forward model. The distinctive features of this algorithm are the usage of a look up table method. The MLS OH concentrations products and N32 Gaussian grid are used to construct the OH concentrations database of four seasons. The observed radiance is obtained by the forward

model. The other factors like spatial information and observation geometries are also simulated according to the characteristic of DSHS for the tomographic retrieval algorithm. The tomographic observed database is established based on the parameters above. The OH concentrations in the target area are obtained through finding in the tomographic observed database directly. The cubic spline interpolation method is also used to obtain the OH concentrations without the corresponding query conditions. The errors are also analyzed caused by the atmospheric mode, the instrument

calibration error, the Doppler effect and interpolation algorithm. The results show that the tomographic retrieval algorithm is not only faster compared with the LSUV retrieval algorithm, but also improves the inversion precision of the OH concentrations especially in the lower atmosphere where the OH concentrations are sensitive.

There are still some problems to be solved when the DSHS will officially work in the future. The performances of two spatial heterodyne spectrometers are inevitably different because they are affected by the temperature, humidity and

electromagnetic environment during manufacturing although the bi-orthogonal structure is theoretically identical in the design parameters. The instrument calibration errors of the instrument are only considered to be 5% at present. This part will be optimized according to the actual working condition after the instrument is officially working.

**Code/Data availability.** All code and data can be obtained from the corresponding author upon request.


**Author contribution.** YA and JM designed the study. YA and GB performed the simulations and carried out the data analysis. JM, WX, and XW provided useful comments on the paper. YA prepared the manuscript with contributions from all co-authors.

**Competing interests.** The authors declare that they have no conflict of interest.

**Acknowledgements.** The authors of this study would like to thank Key Laboratory of Optical Calibration and Characterization of Chinese Academy of Sciences for funding this research. We also would like to thank Dr. Alex Rozanov from Institute of Environmental Physics/Institute of Remote Sensing, University of Bremen, Germany for

helpful advices in using SCIATRAN.

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
