# Peer review of "Tomographic retrieval algorithm of OH concentration profiles using Double Spatial Heterodyne Spectrometers"

_Atmospheric Measurement Techniques, 2019_

## Referee Comment (RC1) · Anonymous Referee #2 · 18 Mar 2020

The authors present a description of the instrument and retrieval algorithm of Double Spatial Heterodyne Spectrometers for the monitor the OH concentration. A forward model is constructed to simulate the observed data of DSHS accurately. A new tomographic retrieval algorithm for obtaining the OH concentrations is also proposed to obtain the OH concentration. The distinctive features of this algorithm are the usage of a look up table method. The errors are also analyzed caused by the atmospheric mode, the instrument calibration error, the Doppler effect and interpolation algorithm. The manuscript demonstrates the potential for measurements of OH concentration by DSHS, but is lacking detail in some areas. Specific comments are provided below.

[Figure]

Abstract: The abstract only describes the method used in the paper. It should give more related results. Page 2, Line 48: The introduction part of the OH radical method should be more summarized, including principle and results. Page 7, line 210: The T2-1 and T2-2 position should be defined more clearly. Page 7, line 212: It should be the distance of satellite position fromT1 to T2-1 moment, and T1 to T2-2 moment. Page 7, line 214: Consider replacing "cannot reflect the actual OH distributions" with "make bigger difference". Page 9, Line 237: Consider making the flow chart more clear and nice looking? Same with other flow charts in the essay. Page 11, Line 297: The "Grating width" in the table 1 should have some unit. Page 11, Line 304: Figure 8 should be replaced by a higher definition picture. Page 17, Line 467: Gave a lots description of the LSUV retrieval algorithm, which was not used in the essay, and did not explain the reason why it was incredible in lower atmosphere. The tomographic retrieval algorithm is an improved version of LSUVïij§Or two independent algorithms? Could you analyse the advantages of the tomographic retrieval algorithm by combining the inversion results of LSUV? Page 18, Line 487: when using the lookup table, the accuracy of the tomographic observed database will affect the OH results? Have you considered proper amount of the data? And the parameter setting of the influencing factors? The accuracy of this database is not described in this paper. And is there any other method to prove the accuracy of the database? Page 20, Line 550: Consider giving more detailed description of the calculation of relative errors. Page 23 line 627: the method of cubic spline interpolation is used to obtain the OH concentration that can not use the lookup table. In essence, the interpolation result is not the actual measurement, and the error estimation of interpolation results in this paper may be inaccurate! Page 25, Line 673,677,679,680: How were these errors calculated? Page 27, line 745: When will the DSHS used in the satellite and obtain the actual data for the OH measurement? The only way to verify the feasibility and correctness of the method is the deeply analysis of the obtained data. The writing of the whole essay need to be improved.

[Figure]

---

## Referee Comment (RC2) · Anonymous Referee #1 · 19 Mar 2020

An et al., "Tomographic retrieval algorithm of OH concentration profiles using Double Spatial Heterodyne Spectrometers" presents a novel proposed approach to remotely sensed atmospheric OH concentrations using a pair of heterodyne spectrometers to record the necessary information to retrieve 3D OH fields in the stratosphere and mesosphere. Given the central role of OH as an oxidant in atmospheric chemistry, this method would certainly be a valuable addition to the fleet of atmospheric monitoring satellites in orbit.

However, I do not find that this paper communicates the proposed observing strategy well enough for a rigorous evaluation of its practicality. I therefore recommend significant revisions and a second round of review before the paper is accepted for final publication.

**General concerns**

- There are many points in this paper where the authors assert a value (e.g. for uncertainty) or conclusion (e.g. the tomographic retrieval is faster than the iterative retrievals) without providing a citation, clear chain of logic, or data to support those points. I list as many as I could find in the remainder of this document. These points need to be addressed so that readers can make their own assessment of these claims.
- What are the plans to test this algorithm in practice once the DSHS is operational. Balloon flights? Aircraft flights? Including this as a future direction would make clear what the next step is.

**Specific comments**

**Sect. 2.2**

Section 2.2 in this paper is intended to outline the observing strategy that this proposed instrument will follow in order to obtain spectroscopic information needed to infer the 3D OH fields. This is a key point that must be communicated clearly for the remote sensing community to evaluate whether this instrument is feasible. However I found this section extremely difficult to read.

The main problem is it seems the authors are exploring different observing strategies and rejecting ones that will not accomplish their goals. It is not clear that this is what they are doing, and it is not clear how they determined whether a particular strategy met their requirements, or even what those requirements are. I recommend this section be completely rewritten to improve its clarity, addressing the following points in order:

- First, clearly explain the criteria that a particular observing strategy must meet to be considered successful
- Second, explain how the strategies were evaluated to determine if they met these criteria
- Third, describe the *final* strategy that meets the necessary criteria

• Only then describe the alternate strategies considered and why they were rejected, making it absolutely clear that these are rejected strategies.

As an example of what makes this confusing, I point to lines 184–193:

The satellite positions and observation geometries are given at the time of T1. **The satellite positions at the time of T2** need to calculate for realizing the three-dimensional segmentation of the target atmosphere. It indicates that there are two satellite positions where the scanning directions of T2-SHS2 and T1-SHS1 are orthogonal to each other in the intersection region at the time of T2 based on the theory mentioned above. Figure 4 shows that there is a tangent point which is defined as P1 when the DSHS detects the atmosphere by the three-dimensional limb mode. The intersection region along the lines of sight of SHS1 (the blue line in the Fig. 4) and SHS2 are the observed data at the time of T1. **There will be two lines of sight (the red lines in the Fig. 4)** where the scanning direction of SHS2 is orthogonal to the SHS1s' under the premise that the working altitude of the satellite is constant. The two lines of sight at the time of T2 (T2-1 and T2-2) and a single line of sight at the time of T1 form two intersection regions R1 and R2 which are symmetrical about P1 along the line of sight at the time of T1.

It is not clear from this section that T2-1 and T2-2 are hypothetical positions, and the bolded text makes it sound like these are actual times when the satellite will make measurements.

I understand that, to some extent, it is necessary to explore these hypothetical options for different measurement times to explain the criteria (e.g. line 194, "That means, the smaller distance between the P1 and R1 (or R2) is, the higher spatial resolution will be."). However, as written, it is unclear when hypothetical options are being considered and when the authors are describing the final option that is chosen.

Some more specific points of confusion:

- p. 5, l. 141: "The hierarchical detection of a series of observed radiances in the same target area comes true through the movement of the satellite platform." Does "movement" refer to the satellite flying along its orbital path, or actively rotating to point the spectrometers at the target area?
- p. 5, l. 139–140 vs. Fig. 3: The description of the two sensors as scanning along and across the satellite's orbit implies, to me at least, that the main axes of the telescopes are pointing parallel to the satellite's direction of flight, but Fig. 3 looks like the sensors are pointing to the right of the direction of flight.
  - This is exacerbated by Table 3, which seems to indicate that there are different T2 points for different altitudes in the profile. Addressing the first point about describing how the satellite moves will help with this.
- p. 6, l. 154–170: Since it's not clear how the satellite will move, it's very hard to understand what's going on here.

- How long will the spectrometers record at T1 and T2? Is it very briefly, or is the spectrometer recording as it sweeps across the target zone?
- Perhaps more generally, what is happening between T1 and T2? Is the satellite reorienting to get the appropriate pointing vector for the measurement at T2? Are the spectrometers on during this time?
- Do I understand right that the point of Fig. 3b is that one cannot use data collected at only one time to retrieve the target area? If so, why not? That is not explained.
- It's not obvious to me why the stratgy shown in Fig. 3c of using SHS1 at T1 and SHS2 at T2 is better than using both SHS1 and SHS2 at both T1 and T2. Surely having SHS1-T1 + SHS2-T1 + SHS1-T2 + SHS2-T2 would provide a better constraint on the OH fields?
- In Fig. 4, why are there red boxes (presumably representing the satellite) on both ends of the T2-x lines of sight but not the T1 line of sight?

**Retrieval algorithm (sect. 4.2)**

Given that the tomographic retrieval described in this section is the major advance in this paper, more detail on how the retrieval works is necessary to fully evaluate it. Specifically, in Fig. 16: what precisely is meant by "SHS1/SHS2 observed data at time of T1/T2"—radiance at a single wavelength, a few selected wavelengths, or all wavelengths measured by the spectrometer?

Presumably it is not a single wavelength because that would make it extremely difficult to disentangle the effects of the background radiance from that of OH fluorescence. But if it is at all measured wavelengths, that seems like it would result in an *extremely* high dimensionality to deal with in the interpolation.

Then, in the first paragraph of sect. 5.1, the authors state that there are four databases, one per season. How are transitions between seasons handled? If it's an abrupt switch (say May 31st uses the spring database and June 1st uses the summer database), won't that introduce discontinuities to the retrieved OH time series?

For Fig. 17, how does the inversion result compare with the true profile that was used to simulate the radiances? Similarly, on line 570, the sentence "The OH concentrations obtained by the LSUV retrieval algorithm are credible in the upper atmosphere," is extemply vague. What level of agreement do you consider "credible"? How large a sample size did you use to determine this?

**Other major concerns**

• Line 580: I don't quite understand "the undisturbed parts are taken as the actual parameters, and the results based on these parts are the true OH fluorescence emission radiance. Then, some certain amounts as the disturbances are applied to the three parts above. The OH fluorescence emission radiance based on these parts is the incorrect results." What are the "undisturbed parts"? Do you mean that you take the values

of each of the atmospheric model, Doppler effect, and instrument calibration from the actual retrieval, and add some error to one at a time? That could be much clearer. How do you determine how much error to add? Do you do this for every OH concentration in the test set or a subset?

- line 597: "...the amount of ozone profile for the disturbances is set as  $\pm 30\%$ ." Why 30%?
- Line 652: "We assumed the instrument calibration errors are  $\pm 5\%$ ..." Why 5%? Please provide either a citation or the line of reasoning used to arrive at this value.
- Line 657: "It is about 0.32% for the six-dimensional cubic spline interpolation of OH concentrations, date parameters, longitude, latitude, solar zenith angle and azimuth angle after research and related experiment." What experiment did you do to determine the 0.32% value?
- Line 741: "The speed of traditional iterative retrieval algorithms is slow and will cost a lot of time." Can you quantify this? How much faster is the tomographic algorithm for, say, a single target area or a day's worth of data?

**Technical corrections**

**Points for clarification**

- Lines 252–255: What is meant by "The SCIATRAN absorption spectrum database is completed based on the HITRAN 2014 database...." Does this mean that the forward model kept the HITRAN 2012 database, but added the UV absorptions from HITRAN 2014? Or does the forward model use HITRAN 2014 instead of HITRAN 2012? Or something else?
- In Eq. (10) where does N(j) come from, if  $N_g(j)$  is either from the OH database or the iterative update?
- Is Section 5.2.4 where we switch from discussion errors in the OH radiance to errors in the actual OH concentration? Please make it clearer that we are transitioning if so; the section title could be "Analysis of total errors in the OH concentrations" and add a first sentence like "We now move from considering errors in the OH radiances to errors in the retrieved concentrations."
- Line 704: "It is a great improvement to compare with the results of the LSUV retrieval algorithm which has a characteristic of iteration." What are the LSUV errors, or where in the paper are they listed? I found some of them later in the paragraph (line 715 and on), but why are they not in Table 7 for immediate comparison with the tomographic errors?
- Line 709: "The corresponding accuracy ranges of MLS OH concentrations products are also given, and its useful height ranges are from 23 to 81 km." Where are these values given?

**Typographic comments**

- line 142: "hierarchical detection," not sure what is meant by this.
- line 228: "The observed data received by the DSHS with an ultra-high spectrum resolution makes up by two parts in this research..." does this mean that the data recorded by the spectrometers will **be made up of** two parts?
- line 247: Citation for the Lifbase software needed
- line 248: Citation for the Bremen atmospheric model needed
- lines 236–275: there are several different ideas in one paragraph (the solar spectrum, the OH spectrum, the OH concentrations database). Please give each its own paragraph.
- Eq. (5): units?
- line 429: "However, it cannot be done to obtain OH concentrations in a converse way directly"—should "converse" be "inverse"?
- Around line 575: it seems like there is a shift from comparing the iteration and lookup table methods to the formal error analysis of the lookup table method. This should be more clearly separated.
- line 646: "The distributions of errors have the same tendency, but the relative errors of tomographic retrieval algorithm are smaller than LSUV retrieval algorithm." Does this mean that the distribution of errors due to the Doppler shift is similar between the LSUV and tomographic algorithm? As written, it sounds like the distribution of errors are similar to the relative errors.
- line 647: "These indicate that tomographic retrieval algorithm can obtain more accurate OH concentrations although the Doppler effect cannot avoid" The tomographic algorithm is more accurate than what? The LSUV algorithm? The Doppler effect cannot avoid what? Does this mean the Doppler effect cannot be avoided by either algorithm?
- line 701: "As the Table 7 indicates that the total errors of OH concentrations are increasing as the heights rise." Since the errors only increase with altitude at first, perhaps instead: "As Table 7 indicates, the total errors of OH concentrations **initially** increase with height..."

---

## Short Comment (SC2) · 26 Mar 2020

Hydroxyl OH is very important for humans to understand the chemical composition of the mesospheric atmosphere, which play an initial role in atmospheric photochemical reactions and initiate the entire oxidation chain in the atmosphere. Due to the limitation of spectral resolution and signal-to-noise ratio of detection technology, there are relatively few OH radical detection devices in the upper and middle atmosphere on board. In this paper, the tomographic retrieval algorithm of OH concentration profiles using Double Spatial Heterodyne Spectrometers has been studied. In order to improve the spatial accuracy of OH radical detection, a spectrometer with higher spectral resolution

than SHIMMER has been developed. The spectrometer is also designed with double SHS and has on-orbit limb observation. In order to verify the detection ability of the hyper-resolution DSHS spectrometer for OH radical, including limb view separation, spectral resolution and other specifications, a number of simulation and Inversion experiments were designed. Firstly, DSHS Spectrometer layout and observation mode were introduced. Next, the forward model based on SCIATRAN was designed. Finally, in order to evaluate the detection ability of weak atmospheric background radiation, the experiment of inversion was carried out, and OH concentration profile data was obtained. The manuscript systematically introduces a new three-dimensional atmospheric detection technology, which is significance for the future development of middle and upper atmosphere payloads. Specific suggestion is provided below. A major issue with this targeted measurement is the increasing Rayleigh scattering for decreasing altitudes (especially in the lower stratosphere). While the manuscript shows a simulated set of interferograms in Figure 8, it does not show that the expected signal to noise ratios will allow the retrieval of OH throughout this altitude range.

---

## Author Response (AR1)

Dear Anonymous Referee #2,

Thank you for your hard work and constructive comments on "Tomographic retrieval algorithm of OH concentration profiles using Double Spatial Heterodyne Spectrometers". The helpful comments have substantially improved our paper. We agree with points raised and modify them in the revised version of the manuscript.

On behalf of the authors
Kind regards,

Yuan An

**A Point-by-Point response to the reviewer's comments**

**Reviewer:** Abstract: The abstract only describes the method used in the paper. It should give more related results.
**Reply:** We agree with your points. Some related results are given in the revised version of the manuscript.

**Reviewer:** Page 2, Line 48: The introduction part of the OH radical method should be more summarized, including principle and results.
**Reply:** We agree with your points. The principle and results of OH radical methods have been summarized. We hope the entire modification is clearly described in the revised version of the new manuscript.

**Reviewer:** Page 7, line 210: The T2-1 and T2-2 position should be defined more clearly.
**Reply:** We apologize for the unclear expression about the T2-1 and T2-2 position. There are two T2 positions which are meeting the requirements of three-dimensional limb mode with the T1 position. We defined the satellite position on the same side to the time of T1 is defined as the position at the time of T2-1, and on the opposite side to the time of T1 is defined as the position at the time of T2-2.

**Reviewer:** Page 7, line 212: It should be the distance of satellite position fromT1 to T2-1 moment, and T1 to T2-2 moment.
**Reply:** We apologize for the unclear expression about these parts in the question. We change the sentence in question to "The distance of the satellite position at the time of T2-2 from the time of T1 is farther than the satellite position at the time of T2-1…" for more accurate expression.

**Reviewer:** Page 7, line 214: Consider replacing "cannot reflect the actual OH distributions" with "make bigger difference".
**Reply:** Changed as suggested. Thank you very much! We change "…cannot reflect

the actual OH distributions…" to "…make bigger difference…".

**Reviewer:** Page 9, Line 237: Consider making the flow chart more clear and nice looking? Same with other flow charts in the essay.
**Reply:** We changed the flow chart as your suggestion. We hope the flow chart is clearer and nicer in the revised version of the manuscript.
**Reviewer:** Page 11, Line 297: The "Grating width" in the table 1 should have some unit.
**Reply:** Apologize for this mistake. The unit "line/mm" has added in the table.
**Reviewer:** Page 11, Line 304: Figure 8 should be replaced by a higher definition picture.
**Reply:** We apologize for this question. Figure 8 has replaced by a new figure which has the resolution of 600 dpi to be clearer.

**Reviewer:** Page 17, Line 467: Gave a lots description of the LSUV retrieval algorithm, which was not used in the essay, and did not explain the reason why it was incredible in lower atmosphere. The tomographic retrieval algorithm is an improved version of LSUV or two independent algorithms? Could you analyze the advantages of the tomographic retrieval algorithm by combining the inversion results of LSUV?
**Reply:** The LSUV retrieval algorithm have been used for many years. The tomographic retrieval algorithm and the LSUV are two independent algorithms. We use LSUV retrieval algorithm to highlight the advantages of the tomographic retrieval algorithm. We hope the entire modification is clearly described in the revised version of the manuscript.

**Reviewer:** Page 18, Line 487: when using the lookup table, the accuracy of the tomographic observed database will affect the OH results? Have you considered proper amount of the data? And the parameter setting of the influencing factors? The accuracy of this database is not described in this paper. And is there any other method to prove the accuracy of the database?
**Reply:** We analyzed the errors of forward model in Sect. 3 and have considered the parameters which affect the OH concentrations. So, we consider the tomographic observed database is accurate.

**Reviewer:** Page 20, Line 550: Consider giving more detailed description of the calculation of relative errors.
**Reply:** Changed as suggested. The descriptions of the calculation of relative errors have been added in the revised manuscript.

**Reviewer:** Page 23 line 627: the method of cubic spline interpolation is used to obtain the OH concentration that cannot use the lookup table. In essence, the interpolation result is not the actual measurement, and the error estimation of interpolation results in this paper may be inaccurate!
**Reply:** Thank you for your comment. Although an effective and reasonable lookup

table inversion algorithm have been constructed, it cannot be obtained the OH concentrations in all cases because 1) the size of lookup table is limited and 2) we cannot consider all situations when the sensor will work in the orbit. The interpolation methods and the threshold judgment methods are always used to solve such problems. A scientific and precise threshold in the threshold judgment method can be obtained according to a lot of OH concentrations data and experiments. However, the OH concentrations data are little now. The interpolation methods are chosen. The cubic spline interpolation method not only has higher stability but also can ensure the continuity and smoothness of the interpolation function under the premise of the ensuring the convergence compared with other interpolation methods like linear interpolation method and some others. The error estimation of the interpolation results is as accurate as possible. We will also improve the accuracy of the OH concentrations when the DSHS work officially in the future.

**Reviewer:** Page 25, Line 673,677,679,680: How were these errors calculated?
**Reply:** We apologize for being unclear about this part. The OH concentrations which are obtained by the LSUV algorithm have been given by Yibo Gao in his graduation thesis and some other papers by researchers. These errors in the question are calculated by the Eq (17) and the error transfer formula. The entire modification is given in the revised version. We hope it is clearly described.

**Reviewer:** Page 27, line 745: When will the DSHS used in the satellite and obtain the actual data for the OH measurement? The only way to verify the feasibility and correctness of the method is the deeply analysis of the obtained data.
**Reply:** We agree to this point deeply. The actual observed data is the key point. The DSHS and research results have been shown and communicated with the related departments. It has been included in the relevant satellite projects for 2020 to 2030.

**Reviewer:** The writing of the whole essay needs to be improved.
**Reply:** Thank you very much for your pertinent suggestion. Our English language do need to be improved. We have found a lot of phrases and grammatical errors through re-reading of the full manuscript carefully. All of the errors have been revised. I hope it is more accurate now on the English expression in the revised paper.

Dear Anonymous Referee #1,
Thank you for your careful work and constructive comments on "Tomographic retrieval algorithm of OH concentration profiles using Double Spatial Heterodyne Spectrometers". The comments have substantially improved our paper a lot. We agree with points raised and modify them in the revised version of the manuscript.

On behalf of the authors
Kind regards,

Yuan An

**A Point-by-Point response to the reviewer's comments**

**General concerns**

**Reviewer:** There are many points in this paper where the authors assert a value (e.g. for uncertainty) or conclusion (e.g. the tomographic retrieval is faster than the iterative retrievals) without providing a citation, clear chain of logic, or data to support those points. I list as many as I could find in the remainder of this document. These points need to be addressed so that readers can make their own assessment of these claims.
**Reply:** Thank you for your suggestions. We have considered your reminders in your document deeply and solved the problems. We hope the entire modification is clearly described in the revised version of the new manuscript.

**Reviewer:** What are the plans to test this algorithm in practice once the DSHS is operational. Balloon flights? Aircraft flights? Including this as a future direction would make clear what the next step is.
**Reply:** We agree to this point. The actual observed data is the key point for the algorithm. Some aircraft flights experiments have been done and the data is being analyzed. The DSHS and research results have been communicated with the related departments. It has been included in the relevant satellite projects for 2020 to 2030. We have added this part to what the next step is.

**Specific comments**

**Sect. 2.2**

**Reviewer:** Section 2.2 in this paper is intended to outline the observing strategy that this proposed instrument will follow in order to obtain spectroscopic information needed to infer the 3D OH fields. This is a key point that must be communicated clearly for the remote sensing community to evaluate whether this instrument is feasible. However, I found this section extremely difficult to read.

The main problem is it seems the authors are exploring different observing strategies and rejecting ones that will not accomplish their goals. It is not clear that this is what they are doing, and it is not clear how they determined whether a particular strategy meet their requirements, or even what those requirements are. I recommend this section be completely rewritten to improve its clarity, addressing the following points in order:

- First, clearly explain the criteria that a particular observing strategy must meet to be considered successful
- Second, explain how the strategies were evaluated to determine if they met these criteria
- Third, describe the final strategy that meets the necessary criteria
- Only then describe the alternate strategies considered and why they were rejected, making it absolutely clear that these are rejected strategies.

As an example of what makes this confusing, I point to lines 184–193:

It is not clear from this section that T2-1 and T2-2 are hypothetical positions, and the bolded text makes it sound like these are actual times when the satellite will make measurements.

I understand that, to some extent, it is necessary to explore these hypothetical options for different measurement times to explain the criteria **(e.g. line 194, "That means, the smaller distance between the P1 and R1 (or R2) is, the higher spatial resolution will be.")**. However, as written, it is unclear when hypothetical options are being considered and when the authors are describing the final option that is chosen.

**Reply:** We agree with your points deeply. We hope the entire modification is clearly described in the revised version of the manuscript especially the work mode of satellite.

Some more specific points of confusion:

**Reviewer:** p. 5, l. 141: "**The hierarchical detection of a series of observed radiances in the same target area comes true through the movement of the satellite platform.**" Does "movement" refer to the satellite flying along its orbital path, or actively rotating to point the spectrometers at the target area?

**Reply:** We apologize for the unclear expression. The "movement" means the satellite fly along its orbital path.

**Reviewer:** p. 5, l. 139–140 vs. Fig. 3: The description of the two sensors as scanning along and across the satellite's orbit implies, to me at least, that the main axes of the telescopes are pointing parallel to the satellite's direction of flight, but Fig. 3 looks like the sensors are pointing to the right of the direction of flight.
– This is exacerbated by Table 3, which seems to indicate that there are different T2 points for different altitudes in the profile. Addressing the first point about describing how the satellite moves will help with this.
**Reply:** We agree with your points. The description about how the satellite moves will help to solve your question. The main axes of the telescopes are pointing parallel to the satellite's direction of flight. Fig. 3 shows that DSHS monitors the OH in the target area using the data of SHS1 at the time of T1 and SHS2 at the time of T2 together. However, the telescopes points to the forward of the direction of flight.

**Reviewer:** p. 6, l. 154–170: Since it's not clear how the satellite will move, it's very hard to understand what's going on here.
**Reply:** We have changed these parts in the revised version of paper. We hope the entire modification is clearly described in the revised version of the manuscript.

**Reviewer:** How long will the spectrometers record at T1 and T2? Is it very briefly, or is the spectrometer recording as it sweeps across the target zone?
**Reply:** The DSHS has the function of hierarchical imaging in the spatial dimensional. It can obtain the data at different heights at the same time without scanning the target area for obtaining the data. So, The DSHS record at T1 and T2 is very briefly.

**Reviewer:** Perhaps more generally, what is happening between T1 and T2? Is the satellite reorienting to get the appropriate pointing vector for the measurement at T2? Are the spectrometers on during this time?
**Reply:** The satellite needs to reorient for getting the appropriate vector for the measurement at the T2 moment. The spectrometers are on during this time.

**Reviewer:** Do I understand right that the point of Fig. 3b is that one cannot use data collected at only one time to retrieve the target area? If so, why not? That is not explained.
**Reply:** Your understanding is correct. If only we use the data at only one time. There are no intersection regions between the field of view slices of SHS1 and SHS2s' at the time of T1. The tomographic data cannot be obtained in this situation.

**Reviewer:** It's not obvious to me why the strategy shown in Fig. 3c of using SHS1 at T1 and SHS2 at T2 is better than using both SHS1 and SHS2 at both T1 and T2. Surely having SHS1-T1 + SHS2-T1 + SHS1-T2 + SHS2-T2 would provide a better constraint on the OH fields?

**Reply:** On the one hand, a lot of data can provide a better constraint but will cause redundancy. The inversion efficiency will reduce. On the other hand, SHS1-T1 + SHS2-T1 + SHS1-T2 + SHS2-T2 will have a lot of noise, it will make the signal-to-ratio low.

**Reviewer:** In Fig. 4, why are there red boxes (presumably representing the satellite) on both ends of the T2-x lines of sight but not the T1 line of sight?
**Reply:** We apologize for the drawing error. The red boxes at the one end of the T2-x lines of sight.

**Retrieval algorithm (sect. 4.2)**

**Reviewer:** Given that the tomographic retrieval described in this section is the major advance in this paper, more detail on how the retrieval works is necessary to fully evaluate it. Specifically, in Fig. 16: what precisely is meant by "**SHS1/SHS2 observed data at time of T1/T2**"—radiance at a single wavelength, a few selected wavelengths, or all wavelengths measured by the spectrometer?
Presumably it is not a single wavelength because that would make it extremely difficult to disentangle the effects of the background radiance from that of OH fluorescence. But if it is at all measured wavelengths, that seems like it would result in an extremely high dimensionality to deal with in the interpolation.
Then, in the first paragraph of sect. 5.1, the authors state that there are four databases, one per season. How are transitions between seasons handled? If it's an abrupt switch (say May 31st uses the spring database and June 1st uses the summer database), won't that introduce discontinuities to the retrieved OH time series?
For Fig. 17, how does the inversion result compare with the true profile that was used to simulate the radiances? Similarly, on line 570, the sentence "**The OH concentrations obtained by the LSUV retrieval algorithm are credible in the upper atmosphere,**" is extremely vague. What level of agreement do you consider "credible"? How large a sample size did you use to determine this?
**Reply:** Thank you for your comments. 1) The SHS1/SHS2 observed data at time of T1/T2 mean the radiance at all wavelengths measured by the DSHS. 2) We count a large number of samples in each grid in the day. It shows that the number of samples is not uniform in the grid, the number of samples in most grids is too small, even there is no profile data in some grids. The less profile data will order to a larger random error. Averaging method cannot reach the goal of smoothing profile and reducing errors. Therefore, the month is determined as the time resolution. We average the profile data in the grid to obtain the monthly average concentration profile data. So, it will not introduce discontinuities to the retrieved OH time series when an abrupt switch happens. 3) The OH concentrations obtained by the tomographic retrieval algorithm are the results of inversion process. The OH concentrations used to simulate the radiance are the parameter of forward process. These OH concentrations have different meaning and are not comparable. The conclusion of results obtained by the LUSV has been proven by a lot of manuscripts. We have added some citations in the revised version of the paper.

**Other major concerns**

**Reviewer:** Line 580: I don't quite understand "**the undisturbed parts are taken as the actual parameters, and the results based on these parts are the true OH fluorescence emission radiance. Then, some certain amounts as the disturbances are applied to the three parts above. The OH fluorescence emission radiance based on these parts is the incorrect results.**" What are the "undisturbed parts"? Do you mean that you take the values of each of the atmospheric model, Doppler effect, and instrument calibration from the actual retrieval, and add some error to one at a time? That could be much clearer. How do you determine how much error to add? Do you do this for every OH concentration in the test set or a subset?

**Reply:** Apologize for these unclear parts in your questions above. Your understanding is exactly what I want to express. We take the values of the atmospheric model, Doppler effect and instrument calibration from the actual retrieval process. Some explanations are given in the following parts to solve the problems about how much errors we add. We do this for every OH concentration in the test subset because the conclusion is representative.

 - line 597: "**...the amount of ozone profile for the disturbances is set as ±30%.**" Why 30%?

**Reply:** We referred to the relevant parameters of the instrument likes SHIMMER which is similar to the DSHS in this part due to the DSHS has not been worked officially. Apart from this, we did a lot of experiments according to the conclusions of the research of SHIMMER and found the 30% is the best threshold to disturb the ozone profile for analyzing the influence of the atmospheric model.

**Reviewer:** Line 652: "**We assumed the instrument calibration errors are ±5%...**" Why 5%? Please provide either a citation or the line of reasoning used to arrive at this value.

**Reply:** We apologize for this unclear description. The instrument calibration error is given by the manufacturing and design department of sensor. Some citations have been added in the revised version of the paper.

**Reviewer:** Line 657: "**It is about 0.32% for the six-dimensional cubic spline interpolation of OH concentrations, date parameters, longitude, latitude, solar zenith angle and azimuth angle after research and related experiment.**" What experiment did you do to determine the 0.32% value?

**Reply:** Apologize for the missing the experiments. We give a certain disturbance to different parameters to calculate the effect of the interpolation algorithm on the OH concentrations.

**Reviewer:** Line 741: "**The speed of traditional iterative retrieval algorithms is slow and will cost a lot of time.**" Can you quantify this? How much faster is the tomographic algorithm for, say, a single target area or a day's worth of data?

**Reply:** The traditional iterative retrieval algorithm will take several or even tens of hours to obtain OH concentrations due to the complex iterative process. The tomographic algorithm just needs minutes for obtaining the OH concentration due to the usage of the lookup table method. It is difficult to quantify how much fast is the tomographic algorithm absolutely. The results obtained by the traditional iterative retrieval algorithm in the lower atmosphere are unscientific but the OH concentrations obtained by tomographic algorithm in the same area are accurate. The tomographic algorithm can improve the accuracy of data while acquiring OH concentrations quickly.

**Technical corrections**

**Points for clarification**

**Reviewer:** Lines 252–255: What is meant by "**The SCIATRAN absorption spectrum database is completed based on the HITRAN 2014 database....**" Does this mean that the forward model kept the HITRAN 2012 database, but added the UV absorptions from HITRAN 2014? Or does the forward model use HITRAN 2014 instead of HITRAN 2012? Or something else?

**Reply**: We apologize for these imperfect descriptions. The SCIATRAN uses HITRAN 2012 by default. However, the HITRAN 2012 database does not include the OH absorption profiles in the ultraviolet band. We use the OH absorption profiles in the HITRAN 2014 to solve this problem. We upgraded the SCIATRAN in this part by using the HITRAN 2014.

**Reviewer:** In Eq. (10) where does N(j) come from, if Ng(j) is either from the OH database or the iterative update?

**Reply:** We apologize for these unclear descriptions. The N(j) is the result of iteration. It can get though the $y_j$ and $N_g(j)$ when the converges meet the accurate requirement or the number of iterations exceeds the iteration number threshold.

**Reviewer:** Is Section 5.2.4 where we switch from discussion errors in the OH radiance to errors in the actual OH concentration? Please make it clearer that we are transitioning if so; the section title could be "**Analysis of total errors in the OH concentrations**" and add a first sentence like "**We now move from considering errors in the OH radiances to errors in the retrieved concentrations.**"

**Reply:** Changed as suggested. Thank you very much! We have changed the title of Section 5.2.4 and added a first sentence "The retrieved results must be considered after the discussion of the OH radiance".

**Reviewer:** Line 704: "**It is a great improvement to compare with the results of the LSUV retrieval algorithm which has a characteristic of iteration.**" What are the LSUV errors, or where in the paper are they listed? I found some of them later in the paragraph (line 715 and on), but why are they not in Table 7 for immediate comparison with the tomographic errors?

**Reply:** The OH concentrations obtained by the LSUV algorithm in the lower atmosphere are unsuitable for scientific research. These limit the applicability of LSUV algorithm. This manuscript focuses on the new algorithm, we use the table to emphasize the errors of new algorithm and consider it is more reasonable to use the text for comparing two algorithms.

**Reviewer:** Line 709: "**The corresponding accuracy ranges of MLS OH concentrations products are also given, and its useful height ranges are from 23 to 81 km.**" Where are these values given?

**Reply:** These values are given by the MLS data quality and description document which seems like the instructions of the data. It is given by the Jet Propulsion Laboratory to ensure the scientificity and accuracy.

**Typographic comments**

**Reviewer:** line 142: "**hierarchical detection,**" not sure what is meant by this.

**Reply:** We apologize for the unclear description about "**hierarchical detection**". The DSHS will detect the altitudes from 15 to 85 km by the three-dimensional limb mode. This observation mode can obtain the OH data at the altitude of 15, 17, 19, 21,23 and the altitudes which follow this role. The atmosphere is divided into many layers. The hierarchical detection means the sensor can obtained the data in these layers like the multi-angle method.

**Reviewer:** line 228: "**The observed data received by the DSHS with an ultra-high spectrum resolution makes up by two parts in this research...**" does this mean that the data recorded by the spectrometers will be made up of two parts?

**Reply:** Correct. The data recorded by the DSHS is made up by two parts due to the ultra-high spectral resolution of spatial heterodyne spectroscopy. The atmospheric background radiance and the OH fluorescence emission radiance can be separated accurately for the following research.

**Reviewer:** line 247: Citation for the Lifbase software needed

**Reply:** Thank you for your suggestion about missing the citation for the Lifbase. The citation has been added in the revised version of the paper.

**Reviewer:** line 248: Citation for the Bremen atmospheric model needed

**Reply:** Thank you for your suggestion about missing the citation for the Bremen atmospheric model. The citation has been added in the revised version of the paper.

**Reviewer:** lines 236–275: there are several different ideas in one paragraph (the solar spectrum, the OH spectrum, the OH concentrations database). Please give each its own paragraph.

**Reply:** We agree with your points. These ideas have been separated clearly. We hope the entire modification is described in the revised version of the paper.

**Reviewer:** Eq. (5): units?

**Reply:** Apologize for missing the units. The Eq. (5) is the wavelength calibration equation. The $\delta_i$ is the symbol of intensity. So, the unit is $phot^{-1}s^{-1}cm^{-2}nm^{-1}sr^{-1}$.

**Reviewer:** line 429: "**However, it cannot be done to obtain OH concentrations in a converse way directly**"—should "converse" be "inverse"?

**Reply:** Apologize for this vocabulary mistake. We have changed the "converse" to "inverse" in this sentence.

**Reviewer:** Around line 575: it seems like there is a shift from comparing the iteration and lookup table methods to the formal error analysis of the lookup table method. This should be more clearly separated.

**Reply:** We agree with your points. These two parts in your question have been separated clearly in the revised version of paper. We hope the entire modification is clearly described.

**Reviewer:** line 646: "**The distributions of errors have the same tendency, but the relative errors of tomographic retrieval algorithm are smaller than LSUV retrieval algorithm.**" Does this mean that the distribution of errors due to the Doppler shift is similar between the LSUV and tomographic algorithm? As written, it sounds like the distribution of errors are similar to the relative errors.

**Reply:** Correct. The distribution of errors is similar between the LSUV and tomographic algorithm and is similar to the relative errors.

**Reviewer:** line 647: "**These indicate that tomographic retrieval algorithm can obtain more accurate OH concentrations although the Doppler effect cannot avoid**" The tomographic algorithm is more accurate than what? The LSUV algorithm? The Doppler effect cannot avoid what? Does this mean the Doppler effect cannot be avoided by either algorithm?

**Reply:** We apologize for being unclear about this. The OH concentrations which are obtained by the tomographic algorithm are more accurate than the results are obtained by the LSUV algorithm. The influence caused by the Doppler effect cannot avoid when the spectral resolution is extremely high when the sensors use the spatial heterodyne spectroscopy technology. So the Doppler effect can be avoided by some situations.

**Reviewer:** line 701: "**As the Table 7 indicates that the total errors of OH concentrations are increasing as the heights rise**." Since the errors only increase with altitude at first, perhaps instead: "**As Table 7 indicates, the total errors of OH concentrations initially increase with height...**"

**Reply:** Changed as suggested. Thank you very much.

Dear Hailiang Shi,

Thank you for your comments on the 'Tomographic retrieval algorithm of OH concentration profiles using Double Spatial Heterodyne Spectrometers'. Your comments have substantially improved our paper.

The increasing Rayleigh scattering as the atmospheric background radiance is the major issue for the lower stratosphere. It is mainly subjected by the function of Rayleigh scattering, ozone absorption and OH self-absorption in this research. We add a table to show the intensity of simulated observation radiance and atmospheric background radiance at some tangent heights which are calculated by the modified SCIATRAN radiative transfer model in the Sect. 3. The errors of results are also given. The OH fluorescence emission radiance can be calculated by subtracting the atmospheric background radiance from the observation radiance. The errors of the inversion results which are given in the Sect. 5.2.4 show the data allow to inverse the accurate OH profile from 15 to 85 km.

On behalf of the authors
Kind regards,

Yuan An

**A list of all relevant changes made in the manuscript**

1. Page 1, Line 17: Change "…its…" to "…OH…"
2. Page 1, Line 22: Add "The error of the results obtained by the forward model is ±44.30% in the lower atmosphere such as 21 km height and decreases gradually until the limit of observation altitude ."
3. Page 1, Line 28: Add "The inversion results are given and the errors of them increase as the altitudes rise until about 41 km height then start to decrease. The errors of the inversion results reach the maximum about ±25.03% in the 41 km height and decrease to ±8.09% in the limited observation height. They are also small in the lower atmosphere which are ±12.96% in the 21 km height. In summary,"
4. Page 2, Line 46: Change "The Fluorescence Assay by Gas Expansion (FAGE), the Differential Optical Absorption Spectroscopy (DOAS) and the Chemical Ionization Mass Spectrometry (CIMS) are commonly used to measure the OH concentrations in the actual limited environments (Hard et al., 1984;Mauldin et al., 1998;Perner et al., 1976). In addition, the $^{14}$CO oxidation method, the Scrubbing using the salicylic acid Technique and the Spin Trapping method are used to obtain the OH concentrations in the laboratory for some theoretical researches (Felton et al., 1990;Salmon et al., 2004;Watanabe et al., 1982). Apart from the six physical and chemical methods mentioned above, many researchers used the high-precision spectrum data from the ground-based instruments especially the Fourier Transform Ultraviolet Spectrometer (FTUVS) in the Table Mountain (Cageao et al., 2001;Cheung et al., 2008;Mills et al., 2002)." to "The Fluorescence Assay by Gas Expansion (FAGE) uses the 308 nm excitation mechanism to excite the OH radical continuously for generating the fluorescent signal to establish the relationship between the OH concentrations (Hard et al., 1984). The Differential Optical Absorption Spectroscopy (DOAS) obtain the OH concentrations because the absorption of OH follows the Lambert-Beer absorption law (Perner et al., 1976).The Chemical Ionization Mass Spectrometry (CIMS) collects ions of OH instead of the photons based on OH oxidation to obtain the OH concentrations (Mauldin et al., 1998) .These methods are commonly used to measure the OH concentrations in the actual limited environments and laboratories. In addition, the $^{14}$CO oxidation method uses the $^{14}$CO$_2$ concentration, enrichment coefficient, reaction rate constant and reaction time to get the OH radical concentrations based on the OH oxidation (Felton et al., 1990). The Scrubbing using the salicylic acid Technique uses an acid and its production rate to obtain OH radical concentrations (Salmon et al., 2004). The Spin Trapping method uses electron spin trap and 4-OH-POBN to obtain the OH concentrations in the laboratory for some theoretical researches (Watanabe et al., 1982). Apart from the six physical and chemical methods mentioned above, many researchers used the high-precision spectrum data from the ground-based instruments especially the Fourier Transform Ultraviolet Spectrometer (FTUVS) in the Table Mountain: The OH P$_1$(1) absorption spectrums which are measured by FTUVS are used to invert the OH concentrations (Cageao et al., 2001). The OH Q$_1$(2) absorption spectrums which are measured by FTUVS are used as auxiliary spectrum to improve the accuracy of OH concentrations (Mills et al., 2002). An improve retrieval method is described that use an average method based on spectral fits to multiple lines weighted by line strength and fitting precision for obtain the OH concentrations. (Cheung et al., 2008)."

5. Page 4, Line 155: Change "The hierarchical detection of a series of observed radiance in the same target area comes true through the movement of satellite platform." to "The hierarchical detection of a series of observed radiance in the same target area comes true when the satellite flies along its orbital path."

6. Page 4, Line 157: Add "It means the sensor can obtain the data of each target height like multi-angle method."

7. Page 8, Line 210: Add "satellite"

8. Page 8, Line 211: Add "satellite"

9. Page 8, Line 213: Change "…cannot reflect the actual OH distributions…" to "…make bigger difference…"

10. Page 8, Line 214: Change "However,…" to "Meanwhile,…"

11. Page 9, Line 221: Change the Figure 4

12. Page 11, Line 243: Change "…the atmospheric background radiance and the OH fluorescence emission radiance." to "…the atmospheric background radiance is the result which the solar radiance is subjected to Rayleigh scattering and trace gas absorption and the OH fluorescence emission radiance is subjected to the OH fluorescence emission mechanism."

13. Page 11, Line 247: Change the Figure 6

14. Page 11, Line 255: Change old paragraph to the new paragraph

15. Page 12, Line 261: Change old paragraph to the new paragraph

16. Page 12, Line 264: Add two citations "(Luque et al., 1999; Sinnhuber et al., 2009"

17. Page 12, Line 267: Change old paragraph to the new paragraph

18. Page 12, Line 267: Add "by default"

19. Page 12, Line 268: Change "The SCIATRAN absorption spectrum database is completed based on the HITRAN 2014 database which contains the OH absorption profiles to solve this problem." to "The OH absorption profiles data in the ultraviolet band is added based on the HITRAN 2014 database into SCIATRAN to solve this problem."

20. Page 12, Line 270: Change old paragraph to the new paragraph

21. Page 12, Line 291: Add "The simulated atmospheric background radiance is mainly subjected by the function of Rayleigh scattering, ozone absorption and OH self-absorption. It is calculated based on some parameters such as the spatial, temporal and observation geometries. An observed radiance profile is given as an example in the (27 °N,106 °E) area in the Fig. 7. The intensity of simulated observation radiance and atmospheric background radiance at some tangent heights are also given in the Table 1."

22. Page 13, Line 300: Add a new Table 1

23. Page 14, Line 314: Add two units "(lines/mm)" in the Table 2

24. Page 14, Line 324: Change the Figure 8

25. Page 19, Line 402: Change the Figure 12

26. Page 20, Line 421: Add "The intensity of simulated observation radiance and atmospheric background radiance at some tangent heights are given in the Table 1. The OH fluorescence emission radiance can be calculated by subtracting the atmospheric background radiance from the observed radiance."

27. Page 21, Line 452: Change "…a converse…" to "…an inverse…"

28. Page 22, Line 480: Add "The each parameters which constitute the database are accurate and the errors of them are analyzed in the Table 3. The method of constructing the database is

reasonable."

29. Page 22, Line 485: Change the Figure 15
30. Page 23, Line 491: Add "…at all wavelengths…"
31. Page 23, Line 492: Add "…at all wavelengths…"
32. Page 24, Line 499: Change the Figure 16
33. Page 24, Line 504: Add "The profile data in the database is averaged to obtain the monthly average concentration profile data. So, it will not introduce discontinuities to the retrieved OH time series when an abrupt switch happens"
34. Page 26, Line 534: Change the Figure 18
35. Page 27, Line 554: Add "…,which can get though the $y_j$ and $N_g(j)$ when the converges meet the accurate requirement or the number of iterations exceeds the iteration number threshold,…"
36. Page 28, Line 598: Change "The OH concentrations obtained by the LSUV retrieval algorithm are credible in the upper atmosphere. These results are the same as the results of MAHSRI and SHIMMER (Conway et al., 1999;Englert et al., 2010). However, the OH concentrations in the lower atmosphere such as below 30 km are unsuitable for scientific research. That is the reason why there are no OH concentrations in these regions from the MAHRSI and SHIMMER (Harlander et al., 2002)." to "The OH concentrations obtained by the LSUV retrieval algorithm are credible in the upper atmosphere which are the same as the results of MAHSRI and SHIMMER (Conway et al., 1999;Englert et al., 2010). However, the OH concentrations in the lower atmosphere such as below 30 km are unsuitable for scientific research because the interference factors like water vapor and ozone are too much. A single spatial heterodyne spectrometer with traditional limb mode cannot also obtain enough and high-quality data to invert the OH concentrations that will make the inversion results unscientific especially in the lower atmosphere."
37. Page 28, Line 605: Add "The tomographic retrieval algorithm and the LSUV are two independent algorithms."
38. Page 28, Line 611: Change "…the undisturbed parts are taken as the actual parameters, and the results based on these parts are the true OH fluorescence emission radiance. Then, some certain amounts as the disturbances are applied to the three parts above. The OH fluorescence emission radiance based on these parts is the incorrect results. The relative errors between the true and incorrect OH fluorescence emission radiance is calculated by the Eq. (17):…" to "1) some certain amounts as the disturbances are applied to the three parts above. The OH fluorescence emission radiance which obtained based on these parts is considered as the incorrect results. 2) the original parts are taken as the actual and undisturbed parameters, and the results based on these parts are the true OH fluorescence emission radiance. The relative errors between the true and incorrect OH fluorescence emission radiance is calculated by the Eq. (17):"
39. Page 31, Line 674: Change "The distributions of errors have the same tendency, but the relative errors of tomographic retrieval algorithm are smaller than LSUV retrieval algorithm. These indicate that tomographic retrieval algorithm can obtain more accurate OH concentrations although the Doppler effect cannot avoid." to "The tendency of relative errors is same, but the relative errors of tomographic retrieval algorithm are smaller than LSUV retrieval algorithm. These indicate that tomographic retrieval algorithm can obtain more accurate OH concentrations than the LSUV retrieval algorithm although the errors caused by Doppler effect cannot avoid when the spectral resolution of DSHS is extremely high."

40. Page 32, Line 681: Add "We assumed the instrument calibration errors are $\pm5\%$ in this research after referring to the detector parameters due to the DSHS do not officially work (Song et al., 2009)"

41. Page 32, Line 683: Add "The cubic spline interpolation method not only has higher stability but also can ensure the continuity and smoothness of the interpolation function under the premise of the ensuring the convergence compared with other interpolation methods like linear interpolation method and some others. The errors estimation of the interpolation results is as accurate as possible."

42. Page 32, Line 689: Add "We give a certain disturbance to different parameters to calculate the effect of the interpolation algorithm on the OH concentrations"

43. Page 32, Line 694: Change "Analysis of total errors" to "Analysis of total errors in the OH concentrations"

44. Page 32, Line 695: Add "The errors of retrieved results must be considered after the discussions about the errors of OH radiance."

45. Page 34, Line 730: Change "As the Table 7 indicates that the total errors of OH concentrations are increasing as the heights rise." to "As Table 8 indicates, the total errors of OH concentrations initially increase with the heights rising."

46. Page 34, Line 733: Add "However, the OH concentrations obtained by the LSUV algorithm in the same area are unsuitable for scientific research mentioned in the Sect. 5.2."

47. Page 34, Line 735: Add "The relative errors of the OH concentrations obtained by LSUV algorithm are calculated by the Eq (17) and the error transfer formula mentioned above."

48. Page 34, Line 740: Change "The corresponding accuracy ranges of MLS OH concentrations products are also given, and its useful height ranges are from 23 to 81 km." to "The corresponding accuracy ranges and useful height ranges of MLS OH concentrations products are also given by the MLS data quality and description document."

49. Page 35, Line 773: Add "However, it is difficult to quantify how much fast is the tomographic algorithm absolutely. The traditional iterative retrieval algorithm will take several or even tens of hours to obtain OH concentrations due to the complex iterative process. The tomographic algorithm just needs minutes for obtaining the OH concentration due to the usage of the lookup table method. The results obtained by the traditional iterative retrieval algorithm in the lower atmosphere are unscientific but the OH concentrations obtained by tomographic algorithm in the same area are accurate."

50. Page 36, Line 813: Add "The actual observed data is the key point for this research. Some aircraft flights experiments have been done and the data is being analyzed. The DSHS and research results have been communicated with the related departments. It has been included in the relevant satellite projects for 2020 to 2030."

51. Page 38, Line 873: Add "Luque J, Crosley D R. LIFbase: Database and Spectral Simulation[J]. 1999"

52. Page 38, Line 885: Add "Sinnhuber, B.-M., Sheode, N., Sinnhuber, M., Chipperfield, M. P., and Feng, W.: The contribution of anthropogenic bromine emissions to past stratospheric ozone trends: a modelling study, Atmospheric Chemistry and Physics, 9, 2863–2871, https://doi.org/10.5194/acp-9-2863-2009, 2009."

[revised manuscript text omitted]

It is a great challenge to monitor the OH in the atmosphere because of its low concentrations and strong activity. The Fluorescence Assay by Gas Expansion (FAGE) uses the 308 nm excitation mechanism to excite the OH radical continuously for generating the fluorescent signal to establish the relationship between the OH concentrations (Hard et al., 1984). The Differential Optical Absorption Spectroscopy (DOAS) obtain the OH concentrations because the absorption of OH follows the Lambert-Beer absorption law (Perner et al., 1976).The Chemical Ionization Mass Spectrometry (CIMS) collects ions of OH instead of the photons based on OH oxidation to obtain the OH concentrations (Mauldin et al., 1998) .These methods are commonly used to measure the OH concentrations in the actual limited environments and laboratories. In addition, the $^{14}CO$ oxidation method uses the $^{14}CO_2$ concentration, enrichment coefficient, reaction rate constant and reaction time to get the OH radical concentrations based on the OH oxidation (Felton et al., 1990). The Scrubbing using the salicylic acid Technique uses an acid and its production rate to obtain OH radical concentrations (Salmon et al., 2004). The Spin Trapping method uses electron spin trap and 4-OH-POBN 
[revised manuscript text omitted]

---

## Referee Report (RR1)

**July 7, 2020**

The authors did a fair job responsing to my individual comments in the author response; however, I did not find most of those clarifications incorporated into the paper. I would like to see the clarifications made in the author response incorportated in some form into the paper. This is true even for my previous comments where I gave my interpretation of what was being said in the paper and my interpretation was correct; the fact that I had to ask whether my understanding was correct meant that I found it unclear.

I will list here my previous comments that were not addressed in the paper and indicate whether the comment in the author response was sufficient.

I also strongly recommend that the authors deposit supporting code and data in a public, persistent repository as recommended in the AMT data policy.

If the authors can revise the manuscript to address these remaining points of confusing and meet AMTs standards for reproducibility, then publication is warranted. However, at this time, major revisions are still required.

- Concerning Sect. 2.2, I previously said: "...it seems the authors are exploring different observing strategies and rejecting ones that will not accomplish their goals. It is not clear that this is what they are doing, and it is not clear how they determined whether a particular strategy met their requirements, or even what those requirements are...." This was not addressed in the revision; for example, it is still unclear to me whether the two T2 points are merely hypothetical and one will be chosen for the operational algorithm, or whether both will be used in the operational algorithm. To reiterate my previous suggestion; I strongly recommend the authors revise the section to answer three questions clearly. One, what criteria must an observing strategy meet to be considered successful? Two, how were different strategies evaluated to determine if they met those criteria? Three, which strategies were considered? Fourth, which strategy was selected? The distinction between which strategies are merely being considered and which one was selected must be **extremely** clear. Please reevaluate the structure of this section very carefully; the lack clarity is due to how this section is organized, not only grammatical errors.
- "The description of the two sensors as scanning along and across the satellite's orbit implies, to me at least, that the main axes of the telescopes are pointing parallel to the satellite's direction of flight, but Fig. 3 looks like the sensors are pointing to the right of the direction of flight." I saw no edit to the paper that made this clearer.

- "How long will the spectrometers record at T1 and T2? Is it very briefly, or is the spectrometer recording as it sweeps across the target zone?" I saw no edit to the paper to make this clearer.
- "Perhaps more generally, what is happening between T1 and T2? Is the satellite reorienting to get the appropriate pointing vector for the measurement at T2? Are the spectrometers on during this time?" I saw no edit to the paper making this clearer. I should also expand by question: the authors indicate that the spectrometers are on between T1 and T2. Is that data recorded used for anything—is it for example recording the T1 spectra for a bunch of other target areas?
- Two comments together: "Do I understand right that the point of Fig. 3b is that one cannot use data collected at only one time to retrieve the target area? If so, why not? That is not explained." Also: "It's not obvious to me why the strategy shown in Fig. 3c of using SHS1 at T1 and SHS2 at T2 is better than using both SHS1 and SHS2 at both T1 and T2. Surely having SHS1-T1 + SHS2-T1 + SHS1-T2 + SHS2-T2 would provide a better constraint on the OH fields?"
  - I did not see an edit to the paper that made these clearer.
  - I'm still not clear on why two times and two spectrometers are necessary. Based on other parts of the response, I'm assuming that the two spectrometers are both pointing along the satellite's direction of flight but measuring in orthogonal planes. That is, if we define the direction of flight as the x-axis, the vector 90° to that but still parallel to the Earth's surface as the y-axis, and the vector pointing away from the Earth as the z-axis, then one spectrometer measures in the xy-plane and the other in the xz-plane.
  - In the response, the authors say that, for measurements at one time, there is no intersection between the spectrometers' fields of view. Why not? If they are orthogonal, their planes of view will intersect. I finally found the field of view (2°) listed in the introduction, instead of the instrument design section. That is important information, which I would expect to be included in the same section as the instrument design (Sect 2.2). How far away from the satellite is the target area? Is it close enough that a 2° FoV is too small for any overlap? If so, why is the FoV so small? Or why could the telescopes not be angled to allow for overlap between the spectrometers' FoV in the target area during a single time of measurement?
  - I don't follow the argument made in the reply that having data from both spectrometers at both T1 and T2 would reduce the signal to noise ratio. Why would considering 4 observations with random noise lead to more noise than only 2 observations? Why wouldn't the noise reduce with  $\sqrt{n}$ ?
  - I'm also not convinced by the assertion in the same reply that combining all four observations (both spectrometers at both times) would provide only redundant data. I understand in principle that *if* the data were redundant it would slow down the lookup since it requires interpolating more coordinates. However, if the data

is truly redundant, then that implies that SHS1 at T1 and SHS2 at T2 provides the same data as SHS1 and SHS2 at T1, which the authors claim is insufficient for the tomographic retrieval. Please provide experimental or theoretical evidence in the paper that SHS1 and SHS2 at T1 only provides insufficient data for the inversion, but that SHS1 and SHS2 at both T1 and T2 provides redudant data. Simply asserting this to be the case is not sufficient.

- For section 4.2, thank for for answering my questions. However I need to follow up on several:
  - Since the authors indicated that all wavelengths from both SHS1 and SHS2 are inputs to the look up table, I'm curious how this lookup is handled. Later in the paper, the look up is referred to as a 6-dimensional spline interpolation. But, from Table 5, the 6 dimensions would seem to be latitude, longitude, altitude, SZA, azimuth angle, and season. How are the OH concentrations that correspond to given radiances looked up?

I'm confused because the only way I can see is if the coordinates for the table included the radiance intensity at each wavelength measured. If even 10 wavelengths are measured, then the table would have 20 spectral dimensions (10 per spectrometer) plus the 6 dimensions mentioned above. If there's just five points in each dimension (which seems quite sparse) and each table OH concentration is stored in 4 bytes, then this means the table would be  $5^{26} \times 4$  bytes which is impossibly large. Please elaborate on how exactly the radiance values from SHS1 and SHS2 are included as dimensions in the table, or more generally in the look up process (and please include this in the paper).

- In Sect. 5.2.1, regarding the 30% uncertainty in ozone, please include the details or reference that supports the 30% uncertainty that were mentioned in the response to the paper itself. Additional details or citations that support the assertion that ozone is the only relevant source of uncertainty (and thus the only parameter of the atmospheric model that needs be discussed) should be included.
- In Sect. 5.2.3, thank you for adding the brief description "We give a certain disturbance to different parameters to calculate the effect of the interpolation algorithm on the OH concentrations". Two comments:
  - 1. A bit more detail would help, i.e. how much you perturbed the different parameters (and why that magnitude perturbation was chosen), how many parameters were perturbed at once, etc. I assume "parameters" here are the coordinates of the look up table, so you tested how much the spline fit changed if e.g. all the SZA were increased by 1°?
  - 2. A separate concern to how dependent the spline interpolation is on the coordinates is how well the spline reproduces known points. So a better experiment might be to remove a set of points from the data the spline is fit to (e.g. remove all points of a specific SZA value), fit the spline to the limited dataset, and test how close it

comes to those known points (similar in concept to withholding test data during machine learning training).

- Sect 3: Regarding the HITRAN database, I am still uncertain what is meant. Do you mean (a) you *completely* removed HITRAN 2012 and use HITRAN 2014 for *all* spectroscopy or (b) you retain HITRAN 2012 but added OH UV absorption lines *only* from HITRAN 2014?
- Sect 5.2.4: Please add a proper citation for the MLS data description document when it is mentioned.

---

## Author Response (AR4)

Dear Keding Lu and AMT,

Thanks for your hard work on "Tomographic retrieval algorithm of OH concentration profiles using Double Spatial Heterodyne Spectrometers". We have learned much from two round of reviews. The anonymous reviewers have given a lot of helpful comments for our manuscript. We will continue our research on this interesting topic!

On behalf of the authors
Kind regards,

Yuan An

################################################################################

Dear editor:

We change some parts in the manuscript. Thanks a lot for your careful work.

On behalf of the authors
Kind regards,

Yuan An

################################################################################

**A list of all relevant changes made in the manuscript**

1. Page 1, Line 26: Add an comma "…, and"
2. Page 2, Line 40: Add an comma "…, and"
3. Page 2, Line 42: Change "…and…" to "…,…"
4. Page 2, Line 53: Add an comma "…, and"
5. Page 3, Line 87: Change "…and…" to "…,…"
6. Page 3, Line 100: Add an comma "…, and"
7. Page 11, Line 256: Change "Forward" to "forward"
8. Page 11, Line 259: Add "(Rozanov et al., 2014)"
9. Page 13, Line 300: Add an comma "…, and"
10. Page 13, Line 335: Add an comma "…, and"
11. Page 22, Line 478: Add an comma "…, and"
12. Page 22, Line 482: Add an comma "…, and"
13. Page 32, Line 706: Add an comma "…, and"
14. Page 32, Line 707: Add an comma "…, and"
15. Page 32, Line 709: Add an comma "…, and"
16. Page 35, Line 757: Add an comma "…, and"
17. Page 37, Line 861: Add an comma "…., and"
18. Page 38, Line 886: Add an comma "…., and"
19. Page 38, Line 891: Add an comma "…., and"

20. Page 38, Line 899: Change "California: Annual average 1997-2000, Geophysical Research Letters, 29, 32-31-32-34, https://doi.org/10.1029/2001gl014151," to "California: Annual average 1997-2000, Geophysical Research Letters, 29, 32-1-32-4, https://doi.org/10.1029/2001gl014151,"

21. Page 39, Line 903: Add "Rozanov, V.V., Rozanov, A.V., Kokhanovsky, A.A., and Burrows, J.P.: Radiative transfer through terrestrial atmosphere and ocean: Software package SCIATRAN, Journal of Quantitativ Spectroscopy & Radiative Transfer, 133, 13-71, https://doi.org/10.1016/j.jqsrt.2013.07.004, 2014."

A marked-up manuscript version

[revised manuscript text omitted]